# ONLINE-OPTIMIZED RAG FOR TOOL USE AND FUNCTION CALLING

## ABSTRACT

In many applications, retrieval-augmented generation (RAG) drives tool use and function calling by embedding the (user) queries and matching them to pre-specified tool/function descriptions. In this paper, we address an embedding misalignment issue that often arises in practical applications due to imperfect embedding models or noisy descriptions; such misalignment may lead to incorrect retrieval and task failure. We introduce Online-Optimized RAG, a deployment-time framework that continually adapts retrieval embeddings from live interactions using minimal feedback (e.g., task success). Online-Optimized RAG applies lightweight online gradient updates with negligible per-query latency and requires no changes to the underlying LLM. The method is plug-and-play: it supports both single- and multi-hop tool use, dynamic tool inventories, and $K$-retrieval with re-ranking. We provide a problem-dependent theoretical analysis that quantifies how the method's performance depends on the initialization quality of the embeddings and other related quantities. Across diverse tool-use and document-retrieval scenarios, our Online-Optimized RAG consistently improves tool selection accuracy and end-task success, thus providing a simple, practical path to robust, self-improving RAG systems.

## 1 INTRODUCTION

Modern large language models (LLMs) increasingly rely on retrieval-augmented generation (RAG) (Lewis et al., 2020) to ground responses in external data. In tool-use settings, an agent encodes the user task, retrieves a tool or function (e.g., an API), and executes it: a query is embedded into a vector space and matched against a catalog of tool descriptions that are likewise embedded; the retriever proposes candidates by similarity (often top-$k$), and an executor (a function-calling API or tool wrapper) carries out the selected call (Patil et al., 2024; Qin et al., 2023; Lumer et al., 2024).

However, when the system cannot incorporate domain feedback, RAG can still yield incorrect calls and answers. Retrieval quality degrades whenever the (trained) embedding geometry drifts from the operational environment. For example, such misalignments can arise from (i) noisy or incomplete tool documentation, (ii) outdated or suboptimal embedding models, (iii) shifts in user intent or phrasing relative to training or others. In such cases, semantically related tools may be mapped far apart (or vice versa), causing the retriever to surface the wrong candidate; the downstream LLM is then bottlenecked by what it is given, leading to unnecessary backtracking or failed tasks. Figure 1 shows two examples of the degradation of retrieval performance caused by bad documentation and a poor embedding model. Existing deployments typically freeze embeddings and indices after offline training (Zeighami et al., 2024; Qin et al., 2023; Patil et al., 2024; Li et al., 2023), leaving no principled, low-cost way to repair performance at deployment. While recent work adapts *controllers* at inference time to decide when or how much to retrieve (Asai et al., 2024; Jeong et al., 2024) or tunes top-$k$ and retrieval strategies (e.g., no retrieval / one-shot / multi-step) via multi-armed-bandit or reinforcement-learning approaches (Fu et al., 2024; Tang et al., 2024; Sun et al., 2025), these methods only adjust global hyperparameters but do not update the underlying embedding space and treat the underlying retriever system as fixed. We defer further discussion of related literature to Appendix A.

We introduce *Online-Optimized RAG*, a deployment-time framework that continuously updates retrieval embeddings from online interactions for tool use and function calling. The core idea is

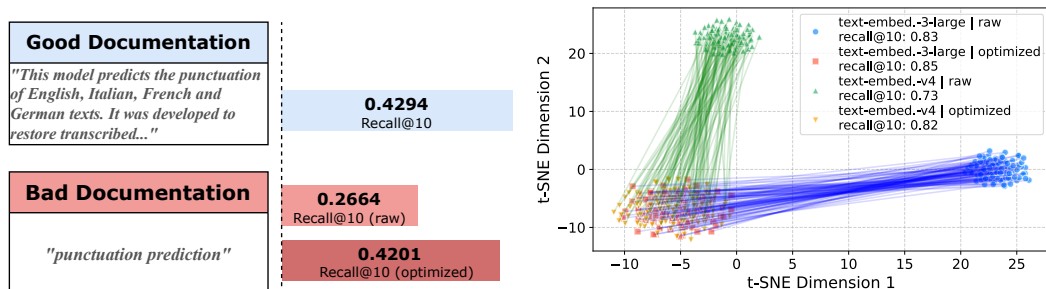

(a) Good documentation vs. bad documentation     (b) Larger model vs. smaller model

Figure 1: We compare the performance between applying Online-Optimized RAG (*optimized*) and without optimization (*raw*). (a) Poor documentation weakens semantic alignment and lowers retrieval quality, while Online-Optimized RAG mitigates this mismatch. (b) t-SNE visualization of the same samples for two embedding models before and after optimization. Initially, `text-embedded-3-large` generally outperforms `text-embedded-v4` and their embeddings are distributed quite differently. However, the embeddings after optimization from both models move toward similar regions and achieve comparable performance, demonstrating our approach's effectiveness. The experimental setup for these two subfigures is provided in Appendix C.

simple: treat the tool retriever as an object to be optimized at test time using minimal observable feedback (e.g., whether the task is solved). After each interaction, we apply lightweight online gradient updates to the item (tool) embeddings to improve future retrieval accuracy without modifying the underlying LLM, planner, or executors. The procedure is plug-and-play, adds negligible latency, can operate under a black-box setting without access to model internals, and also applies beyond tools to general document retrieval. Our contributions are summarized as follows:

**New problem formulation for online retrieval.** We cast the problem of RAG tool and function selection under an online learning framework with bandit-style execution feedback (success/failure signals only for the chosen tool), updating the retrieval geometry on the fly after collecting each feedback. This new formulation itself gives a new perspective to understand RAG tasks.

**A simple, scalable update rule.** We propose an online gradient descent variant that adjusts embeddings per interaction using an importance-weighted estimator. The update of the embeddings keeps computation overhead minimal for large catalogs and high-throughput systems; it is both intuitive and theoretically supported. To the best of our knowledge, this is the first lightweight plug-and-play approach to improve retrieval performance at deployment time through adapting embeddings.

**Versatility across real retrieval settings.** The same update mechanism applies to tool *and* document retrieval, single- and multi-hop pipelines, dynamic tool inventories, and multiple retrievals with reranking. This enables a straightforward integration with common function-calling frameworks and LLM agents without altering the LLM.

**Principled adaptation guarantees.** We derive a problem-dependent performance analysis clarifying how performance depends on the quality of the initial embeddings: strong initializations accelerate convergence toward the optimum, while weaker ones still improve steadily under online updates.

**Empirical evidence on comprehensive tasks.** Across diverse scenarios, our Online-Optimized RAG consistently improves tool selection and downstream task success, and the performance also transfers to general document retrieval. This demonstrates our method as a simple and robust path to self-improving RAG.

## 2  PROBLEM SETUP

### 2.1  RAG FOR TOOL USE AND FUNCTION CALLING

In this section, we describe the problem of RAG and present its mathematical setup. The setup can be viewed as a simplified version of the most general RAG systems. Here we first develop the reader's intuition for our algorithm and introduce the rigorous terminology and notation used for

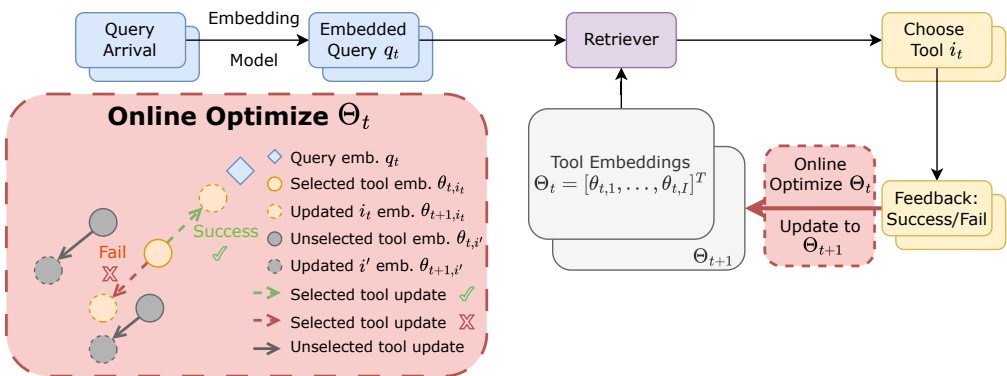

Figure 2: Online-Optimized RAG updates tool embeddings at deployment time for each incoming query $\boldsymbol{q}_t$. When the selected tool $i_t$ succeeds, its embedding moves toward the query to increase similarity. When it fails, the embedding of $i_t$ moves away to reduce similarity. Embeddings of all unselected tools are also pushed away from the query. See Algorithm 1 and its discussion for details.

its formal description. We will then discuss several extensions that cover a broad range of different RAG application contexts.

Specifically, for a RAG system, an embedding model such as OpenAI or Gemini embeddings API, maps an input query (e.g., a prompted question or task for the users) to a query embedding $\boldsymbol{q} \in \mathbb{R}^d$. On the other hand, a database or a tool pool contains $I$ candidate items (e.g., documents for answering the question, functions in MCP, or tools for solving the task). The goal of RAG is to retrieve a proper tool from the $I$ items that best fits the query $\boldsymbol{q}$. For the basic setup, we consider the case that for each query $\boldsymbol{q}$, there exists an (unobserved) optimal item $i^* \in \{1, \ldots, I\}$ that best answers the query or solves the task. The case is motivated by that in most tool-use and function-calling applications, the optimal tool or function is usually unique. However, our online-optimized framework and the algorithm also apply to more general setups of K-retrievals (with rerankers), time-varying database, and multi-hop retrievals which we defer to Section 3.1.

Next, given $\boldsymbol{q}$, the RAG system produces a distribution $\boldsymbol{p} = (p_1, \ldots, p_I)$, where $p_i$ is the probability of selecting item $i$. The cosine similarity-based RAG (most commonly used) represents each item $i$ with an embedding vector $\boldsymbol{\theta}_i \in \mathbb{R}^d$,

$$\boldsymbol{\Theta} = [\boldsymbol{\theta}_1, \ldots, \boldsymbol{\theta}_I]^\top \in \mathbb{R}^{I \times d}.$$

Then each item $i$ is scored by the softmax of the inner product

$$p_i(\boldsymbol{q}, \boldsymbol{\Theta}) = \frac{\exp(\boldsymbol{q}^\top \boldsymbol{\theta}_i)}{\sum_{i'=1}^I \exp(\boldsymbol{q}^\top \boldsymbol{\theta}_{i'})}. \tag{1}$$

In this light, the retrieval problem can also be viewed as a multiclass classification with input $\boldsymbol{q}$ and label $i^*$ under a softmax classifier parameterized by $\boldsymbol{\Theta}$. The loss function is

$$l(\boldsymbol{\Theta}; (\boldsymbol{q}, i^*)) = -\log p_{i^*}(\boldsymbol{q}, \boldsymbol{\Theta}).$$

## 2.2 ONLINE-OPTIMIZED FRAMEWORK FOR RAG

We now present an online-learning setting for the RAG problem where at each time $t = 1, \ldots, T$, a query arrives represented by the embedding $\boldsymbol{q}_t$. Importantly, we allow changing embeddings $\boldsymbol{\Theta}_t \in \mathbb{R}^{I \times d}$ indexed by time $t$. The benefit is that this admits an imperfect initial embedding $\boldsymbol{\Theta}_1$ and allows embeddings to be better learned and improved over time.

In hindsight of seeing all the data $\{(\boldsymbol{q}_t, i_t^*)\}_{t=1}^T$, the optimal embedding should be

$$\boldsymbol{\Theta}^* = \arg\min_{\boldsymbol{\Theta}} \sum_{t=1}^T l(\boldsymbol{\Theta}; (\boldsymbol{q}_t, i_t^*)). \tag{2}$$

In the online setting as how these RAG systems are usually deployed in practice, at each time $t$, we can choose the embeddings $\boldsymbol{\Theta}_t$ based on the past observation history $\mathcal{H}_t = \{\boldsymbol{q}_s, i_s, \mathbb{1}\{i_s = i_s^*\}\}_{s=1}^{t-1}$.

Here $\boldsymbol{q}_s$ is the query embedding at time $s$, $i_s$ is the chosen tool, and the indicator variable tells whether the chosen tool is the correct/optimal one or not. And thus the RAG performance is measured by

$$\sum_{t=1}^{T} l(\boldsymbol{\Theta}_t; (\boldsymbol{q}_t, i_t^*)).$$

We make several important remarks about the setup. First, the online setup is mainly motivated by the sequential arrival of user queries in RAG systems, and the nature makes it possible to continually refine the embeddings $\boldsymbol{\Theta}_t$. The setup also allows a distribution shift of $\boldsymbol{q}_t$ over time, and ideally, $\boldsymbol{\Theta}_t$ should be online optimized to adapt to the shift over time. Second, the feedback structure $\mathbb{1}\{i_t = i_t^*\}$ is mild as it doesn't require knowing the optimal $i_t^*$ but only whether the chosen one $i_t$ equals $i_t^*$ or not. Such a *bandit-style* or *partial-observation* feedback system removes the need for additional data annotations on the optimal $i_t^*$ at each time (which users of the RAG may not even know), but $\mathbb{1}\{i_t = i_t^*\}$ can be simply obtained by users' interactions (thumb-up or -down) or rule-based judges of task success. Third, we choose to optimize the database/toolbase embeddings $\boldsymbol{\Theta}_t$ instead of the query embedding model that gives $\boldsymbol{q}_t$ for two reasons: (a) the query embedding models are sometimes blackbox APIs and don't provide a fine-tuning option, and (b) they are often used simultaneously for other RAG tasks, and fine-tuning against one RAG task may deteriorate its performances on others. Lastly, we note our idea of online-optimizing $\boldsymbol{\Theta}_t$ can be viewed as a lightweight implementation of the tool description rewriting idea in building MCP-based agents (Anthropic, 2025); we optimize in the embedding space, whereas Anthropic (2025) optimizes in the language space, both for the tool use and function callings. In Appendix B.4, we discuss the case where the binary feedback can be noisy.

## 3 ONLINE-OPTIMIZED RAG: ALGORITHMS AND VARIANTS OF RAG

In this section, we present our algorithm of online-optimized RAG and show how it can be applied to several extensions beyond the main setup.

Algorithm 1 implements the standard RAG pipeline when handling a stream of user queries, except for Step 5 and Step 6, where it updates the embeddings $\boldsymbol{\Theta}_t$. Essentially, the update performs a stochastic gradient descent with respect to the loss function equation 2. We note that in calculating the update equation 4, it only requires the knowledge of $\mathbb{1}\{i = i_t^*\}$, i.e., we only need to know whether the chosen item $i_t$ is the correct one or not, but no need to know $i_t^*$. As mentioned earlier, this creates much convenience in annotation – no need for hiring annotators to label $i_t^*$. The most important structural property of the update is described by the following lemma.

**Lemma 3.1.** *For $i = 1, ..., I$,*

$$\mathbb{E}[\boldsymbol{g}_{t,i}] = \left. \frac{\partial l(\boldsymbol{\Theta}; (\boldsymbol{q}_t, i_t^*))}{\partial \boldsymbol{\theta}_i} \right|_{\boldsymbol{\Theta} = \boldsymbol{\Theta}_t}$$

*where the expectation is over the tool selection $i_t \sim \boldsymbol{p}_t$ as defined in Algorithm 1.*

The lemma states that the update term at time $t$ can be viewed as a stochastic gradient of the $t$-th term in the loss function equation 2. This enables a clean theoretical analysis of the algorithm, which we defer to Section 5. The key to achieving this property in Lemma 3.1 is the coefficient before $\boldsymbol{q}_t$. Such a design often appears for bias correction in adversarial online learning (Auer et al., 2002; Kakade et al., 2008).

Intuitively, for the chosen item $i = i_t$, if the choice is incorrect ($i_t \neq i_t^*$), then $\boldsymbol{g}_{t,i_t} = p_{t,i_t} \cdot \boldsymbol{q}_t$ and the update $\boldsymbol{\theta}_{t+1,i_t} = \boldsymbol{\theta}_{t,i_t} - \eta \, p_{t,i_t} \cdot \boldsymbol{q}_t$ moves $\boldsymbol{\theta}_{t,i_t}$ away from $\boldsymbol{q}_t$, decreasing their similarity. If the choice is correct ($i_t = i_t^*$), then $\boldsymbol{g}_{t,i_t} = \left( p_{t,i_t} - \frac{1}{p_{t,i_t}} \right) \boldsymbol{q}_t$, so $p_{t,i_t} - \frac{1}{p_{t,i_t}} \leq 0$ and the update moves $\boldsymbol{\theta}_{t,i_t}$ toward $\boldsymbol{q}_t$, increasing similarity. The magnitude of this correction is proportional to $\left| p_{t,i_t} - \frac{1}{p_{t,i_t}} \right|$, which is larger when the model's current confidence $p_{t,i_t}$ is smaller, i.e., we correct more aggressively when we were unsure yet happened to be right. For all other items $i \neq i_t$, the update $\boldsymbol{\theta}_{t+1,i} = \boldsymbol{\theta}_{t,i} - \eta \, p_{t,i} \cdot \boldsymbol{q}_t$ moves $\boldsymbol{\theta}_{t,i}$ away from $\boldsymbol{q}_t$, decreasing their similarity. This nonzero adjustment for unchosen items is because the loss couples all items through the softmax

normalization, and hence increasing probability on the (unknown) correct item necessarily requires decreasing probability on the others. The dynamics are also visualized in Figure 2.

---

**Algorithm 1** Online-Optimized RAG (ORAG)

---

**Input:** Initial embeddings $\boldsymbol{\Theta}_1 = [\boldsymbol{\theta}_{1,1}, \boldsymbol{\theta}_{1,2}, \ldots, \boldsymbol{\theta}_{1,I}]^\top \in \mathbb{R}^{I \times d}$; learning rate $\eta > 0$

1: **for** $t = 1, 2, \ldots$ **do**
2:     Observe query embedding $\boldsymbol{q}_t \in \mathbb{R}^d$.
3:     Compute sampling probabilities from the current $\boldsymbol{\Theta}_t$ by equation 1:

$$p_{t,i} = p_i(\boldsymbol{q}_t, \boldsymbol{\Theta}_t), \quad i = 1, \ldots, I. \tag{3}$$

4:     Sample an item (tool/document) $i_t \sim \boldsymbol{p}_t = (p_{t,1}, \ldots, p_{t,I})$ and get feedback $\mathbb{1}\{i_t = i_t^*\}$.
5:     Compute the (stochastic) gradient estimate $\boldsymbol{g}_{t,i}$ for each item $i$:

$$\boldsymbol{g}_{t,i} = \left( p_{t,i} - \frac{\mathbb{1}\{i = i_t\}\mathbb{1}\{i_t = i_t^*\}}{p_{t,i_t}} \right) \boldsymbol{q}_t. \tag{4}$$

6:     Update embeddings for $\boldsymbol{\Theta}_{t+1} = [\boldsymbol{\theta}_{t+1,1}, \ldots, \boldsymbol{\theta}_{t+1,I}]^\top$ by

$$\boldsymbol{\theta}_{t+1,i} = \boldsymbol{\theta}_{t,i} - \eta \cdot \boldsymbol{g}_{t,i}.$$

    ▷ *Optional:* project $\boldsymbol{\theta}_{t+1,i}$ into some desired subspace (such as unit ball $\{\boldsymbol{\theta} \in \mathbb{R}^d : \|\boldsymbol{\theta}\|_2 \leq 1\}$)
7: **end for**

---

We make the following remarks about the algorithm:

**Learning rate.** The parameter $\eta$ controls the learning rate of embedding updates. As shown later, a proper choice of $\eta$ yields convergence of the loss toward the optimum. In practice, a small constant (e.g., $\eta = 10^{-5}$) prevents overly large changes. One can also use a time–varying schedule, e.g., $\eta_t = c/\sqrt{t}$ with $c > 0$, to taper updates as more (online) data arrive.

**Session scope.** The algorithm imposes no requirements on how $(\boldsymbol{q}_t, i_t^*)$ are generated. In practice, the embedding updates may aggregate interactions from a broad user population (to adapt universally) or from a single user (to personalize the system). The prototypical version of Algorithm 1 performs updates upon every sample, but one may easily convert it into a batched version by batching samples from multiple timestamps or even an offline version.

**Computation.** As noted earlier, the algorithm is lightweight, and it performs one gradient update at each time step. There is an even more efficient version which only performs an update to the embedding of the chosen item $i_t$. We defer more details to Appendix B.2.

**Exploration.** Unlike Banditron (Kakade et al., 2008) for online multiclass prediction, which enforces uniform exploration with a fixed probability, our procedure utilizes the inherent randomness of the vector $\boldsymbol{p}_t$. The advantage of this exploration-free design is that it doesn't sacrifice the current user experience for future improvement of the system.

### 3.1 VARIANTS OF THE RAG SETUP

Now we show how Algorithm 1 can be applied to more general RAG settings than the setup in the last section. We report its numerical performance in the next section, and defer more implementation details to Appendix B.3.

$K$ **retrievals with reranker.** Algorithm 1 retrieves one single tool/function per round. A practical extension is to retrieve $K \geq 2$ candidates and pass them to a reranker (e.g., a cross-encoder or an LLM judge) that selects the best among them (Qin et al., 2023; Xu et al., 2024). In Algorithm 2, we deal with the RAG system with a reranker; instead of sampling one item, it samples multiple items and lets the reranker decide the best one. The algorithm thus modifies the sampling step of Algorithm 1 (line 4) by inserting a reranking block.

**Time-varying database.** Algorithm 1 is also compatible with a dynamic toolbox $\{1, \ldots, I\}$ that changes over time. In its variant Algorithm 3, at the start of round $t$, it first updates the available set of items and adjusts the embedding matrix $\boldsymbol{\Theta}_t$ accordingly (adds rows for new items and removes rows

for obsolete ones). Then compute $p_t$ and proceed as usual, i.e., ensure $\Theta_t$ contains exactly the items available at time $t$. This operates smoothly because (i) the sampling distribution is softmax-based (it automatically re-normalizes over the current items), and (ii) the updates are item-wise (lines 5-6 of Algorithm 1). This setting captures the case where the optimal tool for certain queries may not exist in early phases (small $t$) and only becomes available at a later stage. For example, the optimal tool $i'$ for a query $q'$ is unavailable for $t < 10$, and $q_t = q'$ for all $t \leq 10$. Even without seeing $i'$, Algorithm 3 will repeatedly push the existing item embeddings away from $q'$ whenever the sampled item is incorrect. This decreases their logits $q'^\top \theta_{t,i}$ and thus their softmax probabilities relative to the (eventual) optimal item. When $i'$ is introduced at $t = 10$, even with an untouched, reasonable initialization aligned to $q'$, its selection probability will be comparatively higher, improving retrieval without any special warm start.

**Multi-hop retrieval.** Some RAG tasks require *multi-hop* retrieval to select multiple items that jointly solve the task (Tang & Yang, 2024). A common strategy is to use a planner (e.g., an LLM) to decompose the input into sub-tasks (Shen et al., 2023; Qin et al., 2023; Lumer et al., 2024). In such a setting, we can apply Algorithm 1 at each hop by reducing the multi-hop query/task to a sequence of single-hop sub-tasks. Concretely, in the variant Algorithm 4, at hop $h$ (the $h$-th sub-task), we run Algorithm 1 to select an item and obtain feedback from a judge (e.g., an LLM or rule-based judge when a human is unavailable) indicating whether the selection advances or answers the query. These per-hop updates align the embeddings across the entire multi-hop pipeline.

Algorithms 2, 3 and 4 are all formally described in Appendix B.3.

## 4 EXPERIMENTS

For Algorithm 1 and its variants, we evaluate them on both tool calling and information retrieval tasks and conduct experiments on several open source benchmarks. We summarize the experiment setup here and defer the implementation details to Appendix C. Unless otherwise noted, all results of our methods are computed as the average of five independent runs.

**Datasets.** For tool use, we adopt *UltraTool* (Huang et al., 2024) and three sub-tasks from *Tool-Ret* (Shi et al., 2025): *ToolRet-Web*, *ToolRet-Code*, and *ToolRet-Customized*. For information retrieval, we use *FiQA* benchmark (Thakur et al., 2021). For multi-hop reasoning, we use *MultiHopRAG* (Tang & Yang, 2024). These datasets provide real-world scenarios for retrieval tasks.

**Baselines.** We compare our method against a strong suite of retrieval models following the methodology in Shi et al. (2025). The baselines include a sparse retriever based on `BM25` (Huang et al., 2024), competitive dense retrievers accessed via API: OpenAI's `text-embedding-3-large` and Qwen's `text-embedding-v4`, and also two state-of-the-art cross-encoder models of different sizes based on previous research and benchmark reports (Muennighoff et al., 2022; Tang & Yang, 2024; Shi et al., 2025): `Qwen3-Reranker-0.6B` and `bge-reranker-v2-gemma`.

**Metrics.** For all retrieval tasks, we report performance using standard information retrieval metrics: Recall@k (`R@k`) and NDCG@k (`N@k`). Following common practice, we choose $k = 10$. For tool-use simulation experiments, we also report the function-call accuracy.

### 4.1 RETRIEVAL PERFORMANCE

We begin by assessing Algorithm 1 through a comparison with strong baselines in the retrieval literature. We report metrics after applying the method with an average of 3000 updates to the embeddings (exact numbers vary based on the selected batch size and dataset size), and we include the initial models without online updates. Table 1 presents the results, where **Ours** denotes the results of Algorithm 1. The results give several key insights. First, our proposed method demonstrates a significant and consistent improvement over its base dense retrieval models. For example, on the ToolRet-Code benchmark, both the `text-embedding-large-3` and the `text-embedding-v4` baselines gain significant performance improvements, and the initially underperformed `text-embedding-v4` even outperforms the `text-embedding-large-3` after the optimization via our method. This shows our approach is not only effective but also versatile, enhancing strong existing models without requiring architectural changes. Second, traditional sparse retrieval methods like `BM25`, which rely on lexical matching, consistently underperform across all

benchmarks. This highlights the necessity of semantic understanding for the nuanced task of tool retrieval, where the user's intent may not share keywords with the tool's description. Finally, while powerful reranker models can achieve high performance on specific tasks, their practical utility is often limited by high computational costs, making them unsuitable for real-time applications. As visualized in Figure 3, our method provides a much more balanced and practical solution, achieving state-of-the-art performance while maintaining low inference latency.

Table 1: Retrieval performance at k=10. The best result in each column is highlighted. Standard errors are given after each score value. Percentage improvements of our methods over their baselines are shown below each score. Dataset names are abbreviated: **U-Tool** (UltraTool), **T-Web** (ToolRet-Web), **T-Code** (ToolRet-Code), and **T-Custom** (ToolRet-Customized).

| Method | U-Tool | | FiQA | | T-Web | | T-Code | | T-Custom | |
|---|---|---|---|---|---|---|---|---|---|---|
| | R@10 | N@10 | R@10 | N@10 | R@10 | N@10 | R@10 | N@10 | R@10 | N@10 |
| BM25 | 0.3208 | 0.2003 | 0.2955 | 0.2326 | 0.1778 | 0.1428 | 0.3446 | 0.2421 | 0.4922 | 0.3816 |
| bge-reranker-v2-gemma | 0.8448 | 0.5852 | **0.7500** | 0.4655 | **0.4849** | **0.3486** | **0.6081** | **0.5322** | 0.6455 | 0.5221 |
| Qwen3-Reranker-0.6B | 0.7200 | 0.4590 | 0.5500 | 0.4361 | 0.3622 | 0.1897 | 0.5802 | 0.4781 | 0.6274 | 0.4923 |
| text-embedding-v4 | 0.7451 | 0.5064 | 0.5335 | 0.4604 | 0.2701 | 0.1453 | 0.5291 | 0.3770 | 0.5066 | 0.4097 |
| text-embedding-3-large | 0.8356 | 0.6067 | 0.6258 | 0.5462 | 0.3243 | 0.1675 | 0.5347 | 0.3582 | 0.6378 | 0.5204 |
| **Ours (text-emb.-v4)** | $0.8256_{\pm.018}$ (+8.05%) | $0.5982_{\pm.011}$ (+8.28%) | $0.5464_{\pm.008}$ (+1.29%) | $0.4698_{\pm.014}$ (+0.94%) | $0.3657_{\pm.036}$ (+9.56%) | $0.1968_{\pm.023}$ (+5.15%) | $0.5960_{\pm.012}$ (+6.69%) | $0.4280_{\pm.016}$ (+5.10%) | $0.5739_{\pm.018}$ (+6.73%) | $0.4398_{\pm.022}$ (+3.01%) |
| **Ours (text-emb.-3-L.)** | $\mathbf{0.8682}_{\pm.018}$ (+3.26%) | $\mathbf{0.6522}_{\pm.013}$ (+4.55%) | $0.6421_{\pm.017}$ (+1.63%) | $\mathbf{0.5680}_{\pm.013}$ (+2.18%) | $0.3780_{\pm.019}$ (+5.37%) | $0.2065_{\pm.017}$ (+3.90%) | $0.5849_{\pm.024}$ (+5.02%) | $0.4070_{\pm.019}$ (+4.88%) | $\mathbf{0.6937}_{\pm.020}$ (+5.59%) | $\mathbf{0.5735}_{\pm.017}$ (+5.31%) |

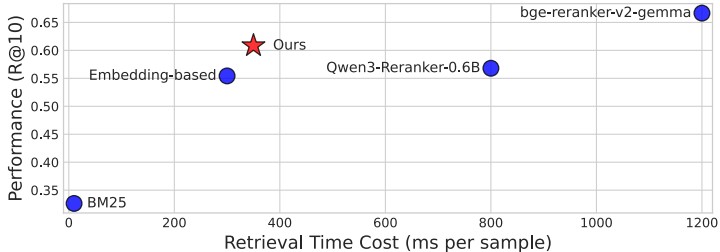

Figure 3: Performance vs. time cost of different retrieval methods. The performance is the arithmetic average of R@10 results in Table 1, and the time cost is evaluated and recorded on the same GPU server. The embedding model time cost uses the `Qwen3-Embedding-4B` as the proxy.

## 4.2 ALGORITHM 1'S VARIANTS EVALUATION

We now evaluate the adaptability of our method across several practical scenarios, including integration with rerankers, time-varying databases, and multi-hop retrieval tasks (see the variants discussed in Section 3.1). We use the *UltraTool* benchmark for experiments on dynamic databases and integration with rerankers, and the *MultiHopRAG* benchmark for the multi-hop retrieval task. The detailed experiment setup is provided in Appendix C.

**Integration with Rerankers.** We consider a pipeline where an LLM reranks the top candidates retrieved by our model before a final tool is selected. For each query, a reranker model reranks the sampled 10 tool documentations, and the success of the final tool call provides the gradients for our algorithm as shown in Algorithm 2. For reproducibility, we employ RankGPT (Sun et al., 2023) with `gpt-4.1-nano-2025-04-14` as the reranker. We compare this LLM-as-reranker approach against our standard method that samples directly from the learned policy and also the baseline, where we make no updates to embeddings. The results are presented in Figure 4. We observe that during the early stage, it is indifferent whether to use a reranker or not. Later, the reranker accelerates improvement in retrieval performance. The reason is that a stronger reranker increases the probability of selecting the correct item. Intuitively, a successful retrieval yields a precise signal that the chosen item is correct, while a failed retrieval only indicates that the chosen item is incorrect without revealing which item is correct. By increasing the rate of successful retrievals, the reranker provides more informative feedback for subsequent learning.

**Time-varying database.** We study a setting where the toolbase changes over time. At the start, only a random subset of tools is available, and the remaining tools are introduced after half of the queries have been processed. Under this setup, part of the embeddings cannot be updated during the first phase, and for some queries, the ground truth optimal tool may be temporarily unavailable. Even so, our method can improve the performance as discussed in Section 3.1. We compare this

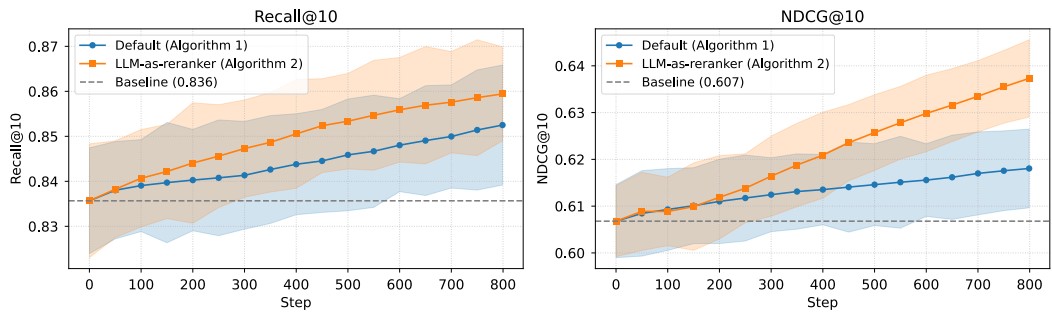

Figure 4: Performance on *UltraTool* with and without an LLM-based reranker. Standard errors are plotted as colored regions. Integrating an LLM reranker provides a stronger signal, accelerating learning and further boosting retrieval performance.

dynamic setting, labeled as *Dynamic DB (Algorithm 3)*, with a static baseline where all tools are available from the beginning, labeled as *Default (Algorithm 1)*. As in Figure 5, though removing embeddings at the beginning can reduce recall, our method adapts to the changing set of tools and achieves consistent gains.

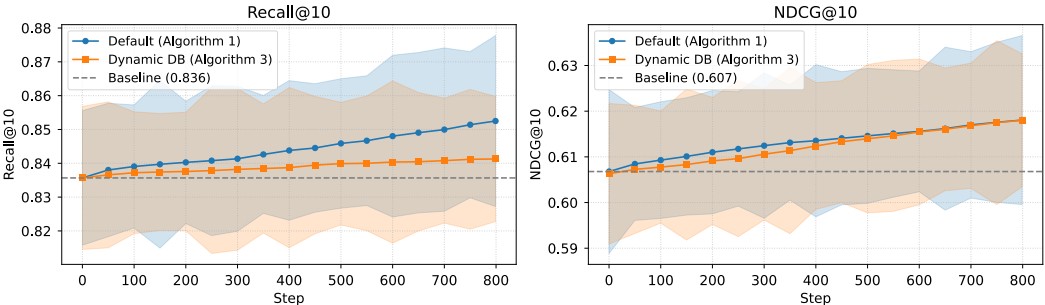

Figure 5: Performance on *UltraTool* for static vs. dynamic database. Standard errors are plotted as colored regions. Our method demonstrates robust adaptation, maintaining consistent improvements even when the toolset changes midway through the experiment.

**Multi-hop retrieval.** We evaluate our method in a multi-hop setting, where solving an input task requires a sequence of successful tool retrievals. The plug-and-play nature of our algorithm enables straightforward integration into the existing multi-hop frameworks. We implement a query decomposition pipeline in which a planner first decomposes the input task into several subtasks, and Algorithm 1 is applied to each subtask, as discussed in Section 3.1. Here, for each subtask query, we retrieve only 5 documents. We evaluate on the *MultiHopRAG* benchmark, and the performance changes are shown in Figure 6. This integration yields a substantial improvement in end-to-end question answering accuracy, from **0.55** to **0.68**.

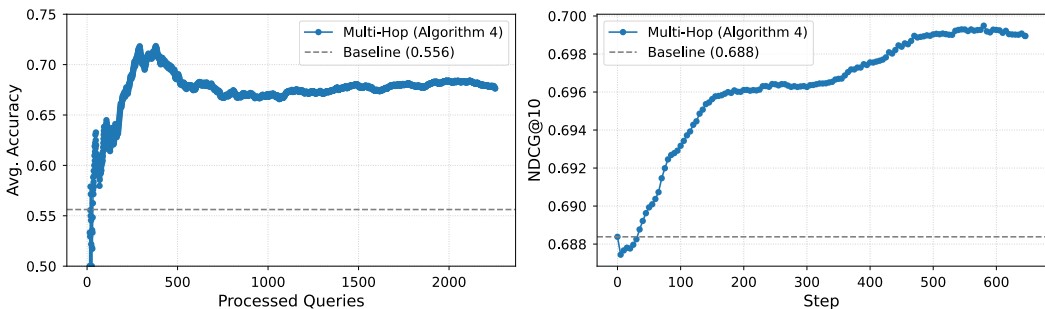

Figure 6: Performance changes on the *MultiHopRAG* benchmark. The baseline is computed using the same retrieval and question-answering workflow (see Appendix C) with raw `text-embedding-3-large` embeddings. Integrating our method into a task decomposition pipeline demonstrates stable learning, leading to improved multi-hop QA performance.

## 5 Theoretical Analysis

In this section, we provide a theoretical analysis of Algorithm 1. The aim is to derive more insights for implementing the algorithm in practice. Generally, the performance of an online algorithm/policy $\pi$ is measured by its *regret*

$$\text{Reg}^\pi\big(\{(\boldsymbol{q}_t, i_t^*)\}_{t=1}^T\big) = \sum_{t=1}^T l(\boldsymbol{\Theta}_t; (\boldsymbol{q}_t, i_t^*)) - \sum_{t=1}^T l(\boldsymbol{\Theta}^*; (\boldsymbol{q}_t, i_t^*)),$$

where $\boldsymbol{\Theta}_t$ is the embedding at time $t$ specified by the policy $\pi$, Algorithm 1 in our context, and $\boldsymbol{\Theta}^*$ is the optimal embedding defined by equation 2 upon optimizing over all the queries in a hindsight manner. As noted earlier, we make no assumption on the generation of $\boldsymbol{q}_t$ and $i_t^*$.

**Theorem 5.1.** *For any sequence* $\{(\boldsymbol{q}_t, i_t^*)\}_{t=1}^T$, *Algorithm 1 (ORAG) with initialization* $\boldsymbol{\Theta}_1$ *and learning rate* $\eta > 0$ *satisfies*

$$\mathbb{E}\big[\text{Reg}^{ORAG}\big(\{(\boldsymbol{q}_t, i_t^*)\}_{t=1}^T\big)\big] \leq \frac{\|\boldsymbol{\Theta}_1 - \boldsymbol{\Theta}^*\|_F^2}{2\eta} + \frac{\eta}{2}\sum_{t=1}^T\left(\frac{1}{p_{t,i_t^*}} - 2p_{t,i_t^*} + 1\right)\|\boldsymbol{q}_t\|_2^2,$$

*where* $\|\cdot\|_F$ *denotes the Frobenius norm and the expectation is w.r.t. the randomness of* $i_t$'s.

Theorem 5.1 gives a problem-dependent regret bound for Algorithm 1. The first term depends on the initialization $\boldsymbol{\Theta}_1$, whereas the second depends on the probabilities $p_{t,i_t^*}$ of selecting the optimal items. For the initialization quality, the term $\|\boldsymbol{\Theta}_1 - \boldsymbol{\Theta}^*\|_F^2$ quantifies how close the initial embeddings are to the optimum. If the initialization is good (i.e., close to $\boldsymbol{\Theta}^*$), only minor updating is needed. For the second term, we can interpret it as the confidence in the optimal item. The summation grows when the model assigns low probabilities to the optimal item. Intuitively, lower confidence (smaller $p_{t,i_t^*}$) incurs larger regret. In particular, if $p_{t,i_t^*} = 1$ then the contribution at time $t$ is zero and the corresponding gradient $\boldsymbol{g}_{t,i}$ vanishes for all $i$ and there is no need to adjust the embeddings. Further, in an unrealistically ideal case, if $p_{t,i_t^*} \equiv 1$ for all $t$, then $\text{Reg}^{ORAG} = 0$ and Algorithm 1 leaves $\boldsymbol{\Theta}_t \equiv \boldsymbol{\Theta}_1$ unchanged. In the light of Lemma 3.1, the proof follows the standard analysis of online gradient descent (Hazan et al., 2016) and is deferred to Appendix D.

With an appropriate choice of $\eta$ to trade off these two aspects, Algorithm 1 achieves sublinear regret in $T$; equivalently, the average regret tends to zero and the loss approaches the optimum of $\boldsymbol{\Theta}^*$:

**Corollary 5.2.** *Assume there exist constants* $\bar{\Theta} > 0$, $\underline{p} \in (0, 1)$, *and* $\bar{q} > 0$ *such that* $\|\boldsymbol{\Theta}_1 - \boldsymbol{\Theta}^*\|_F^2 \leq \bar{\Theta}$, $p_{t,i_t^*} \geq \underline{p}$, *and* $\|\boldsymbol{q}_t\|_2^2 \leq \bar{q}$ *for all* $t$. *Then, with* $\eta = \sqrt{\frac{\underline{p}\,\bar{\Theta}}{\bar{q}\,(1-\underline{p})(1+2\underline{p})\,T}}$, *we have*

$$\mathbb{E}\big[\text{Reg}^{ORAG}\big(\{(\boldsymbol{q}_t, i_t^*)\}_{t=1}^T\big)\big] \leq \sqrt{\frac{\bar{\Theta}\,\bar{q}\,(1-\underline{p})(1+2\underline{p})\,T}{\underline{p}}} = O(\sqrt{T}).$$

Corollary 5.2 shows that Algorithm 1 attains $O(\sqrt{T})$ regret relative to the optimal embeddings (knowing all incoming queries in hindsight). While the choice of $\eta$ above depends on several parameters, in practice (and in our experiments), a small constant with a time-varying schedule, e.g., $\eta_t = c/\sqrt{t}$ with $c = 10^{-5}$ (as used in standard online convex optimizations (Hazan et al., 2016)) can work well across different contexts. Figure 8 in Appendix C empirically verifies a sublinear cumulative regret for Algorithm 1. We also draw a connection between the cross-entropy loss (used in the above regret analysis) and the accuracy metric, and we refer to Appendix B.

## 6 Conclusion

We introduce *Online-Optimized RAG*, a deployment-time framework for tool use and function calling that continually improves retrieval by updating embeddings from live interactions with minimal feedback. Our method casts retrieval as online classification and employs lightweight gradient-style updates that preserve latency and throughput, scale to large catalogs, and integrate seamlessly with existing LLM pipelines without retraining the generator. We provide theoretical guarantees and an analysis linking initial embedding quality to downstream performance, supported by empirical evaluation on real retrieval workloads. We hope this work catalyzes future deployment of self-improving RAG systems.

## REPRODUCIBILITY STATEMENT

To ensure full reproducibility, our source code is provided in the supplementary material. Please note that reproducing experiments involving external LLM APIs will require a valid API key. The original datasets are publicly available on the Hugging Face Hub, with detailed processing steps described in Appendix C.1. We will release the processed datasets upon publication. For our theoretical contributions, detailed proofs and extended discussions are available in Appendix D and Appendix B, respectively.

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

# A RELATED WORK

## A.1 RETRIEVAL AUGMENTED GENERATION

Retrieval-augmented generation (RAG) (Lewis et al., 2020) augments LLMs with a retriever that supplies passages from an external knowledge base and instructs the model to answer using those passages. By exposing sources, RAG reduces the risk of hallucinations and improves factuality. We refer to the survey paper Oche et al. (2025) for a comprehensive review of RAG.

The first line of work adapts when and how much to retrieve at inference time rather than changing the retriever itself. SELF-RAG (Asai et al., 2024) lets the model decide when to retrieve and critique its own outputs, reducing hallucinations over standard RAG, while Adaptive-RAG (Jeong et al., 2024) learns a lightweight router that sends easy questions to zero/single-shot retrieval and harder ones to multi-step pipelines, trading accuracy for latency. Building on online signals, AutoRAG-HP (Fu et al., 2024) frames top-$k$ (and related knobs) as a hierarchical bandit tuned from live feedback; MBA-RAG (Tang et al., 2024) treats whole retrieval policies (none/one-shot/multi-step) as arms; and DynamicRAG (Sun et al., 2025) optimizes a reranker via reinforcement learning to reorder passages and choose $k$ per query. These methods primarily tune controller hyperparameters rather than updating the embedding space as we do.

Another line of work swaps or augments the retrieval substrate itself. Parametric RAG (Su et al., 2025) pre-parameterizes documents as small LoRA adapters so the model retrieves by merging adapters instead of consuming long contexts, while HippoRAG and its follow-up (Jimenez Gutierrez et al., 2024; Gutiérrez et al., 2025) build an open knowledge graph and use graph walks to achieve multi-hop, context-aware retrieval.

Closer to our aim of aligning the retriever with usage signals, several papers adjust representations at inference or through fine-tuning. RAFT (Zhang et al., 2024) fine-tunes generators to quote the right spans, improving faithfulness under noisy top-$k$. For continual retriever training, Goswami et al. (2025) estimates query-embedding drift for new tasks and compensates it at retrieval time to preserve compatibility with an existing index. ReFIT (Reddy et al., 2023) distills a cross-encoder reranker into the query embedding on the fly and re-retrieves with the updated query vector; FLAIR (Zhang et al., 2025) leverages user/synthetic indicator feedback to re-rank via a two-track scoring scheme; and NUDGE (Zeighami et al., 2024) fine-tunes document embeddings through offline training and validation datasets with positive feedback. However, these approaches are controller-tuning, offline, or/and require labeled offline datasets. In contrast, our method performs lightweight online gradient updates to the retrieval embeddings from minimal deployment feedback (e.g., solved/unsolved).

## A.2 TOOL USE AND FUNCTION CALLING

Tool use and function calling are now core capabilities of modern LLMs: models can invoke external resources to complete tasks by calling APIs (Qin et al., 2023; Patil et al., 2024; Li et al., 2023), executing a Python interpreter (Gao et al., 2023), or orchestrating other AI models (Shen et al., 2023). In particular, RAG is commonly employed in the function-call setting for tool-augmented LLMs: given a user query, the system retrieves tool/function specifications and examples from a catalog so the model can select and parameterize the correct call (e.g., Shen et al. (2023); Liu et al. (2025); Lumer et al. (2024); Alazraki & Rei (2024); Xu et al. (2024)).

To strengthen tool use, methods generally combine two phases: offline training and online inference. Offline approaches fine-tune LLMs on curated tool-use corpora (Qin et al., 2023; Patil et al., 2024; Li et al., 2023; Hao et al., 2023; Wang et al., 2024; Schick et al., 2023). Online techniques improve calling performance at inference time by supplying clearer tool descriptions, leveraging the model's reasoning, and incorporating feedback loops (Yuan et al., 2024; Alazraki & Rei, 2024; Lumer et al., 2024; Xu et al., 2024; Shen et al., 2023). Within the feedback-driven line, PEToolLLaMA (Xu et al., 2025) personalizes tool learning through supervised fine-tuning and direct preference optimization, while Xu et al. (2024) iteratively refines queries using tool feedback to improve retrieval accuracy at the cost of additional latency. Both frameworks require offline model updates and/or multi-step inference. By contrast, our approach targets the retrieval layer that underpins function selection: we perform lightweight online gradient updates to the retrieval embeddings using minimal deployment feedback, aligning tool retrieval without fine-tuning the LLM or adding complex controllers. This yields a plug-and-play mechanism for robust, self-improving tool use during deployment.

# B MORE DISCUSSIONS

## B.1 DISCUSSION FOR CROSS-ENTROPY LOSS

We choose the cross-entropy loss since its convexity in $\Theta$ and also its surrogate property for the 0–1 loss (Tewari & Bartlett, 2007; Bartlett et al., 2006): optimizing $\Theta$ by minimizing cross-entropy is statistically aligned with maximizing top-1 retrieval accuracy.

The retrieval task can be cast as a multiclass prediction with input $q$ and label $i^*$. The *Bayesian* 0–1 risk of a decision rule $g(q) \in \{1, \ldots, I\}$ is

$$\mathcal{R}_{0\text{--}1}(g) = \Pr\big(g(\boldsymbol{Q}) \neq I^*\big) = \mathbb{E}\big[\,\mathbb{1}\{g(\boldsymbol{Q}) \neq I^*\}\big],$$

whose Bayes-optimal rule is $g^\star(q) = \arg\max_i \eta_i(q)$, where $\eta_i(q) := \Pr(I^* = i \mid \boldsymbol{Q} = q)$ and the above probability is with respect to the randomness of $(\boldsymbol{Q}, I^*)$. Directly minimizing the 0–1 risk is intractable; a standard approach is to minimize the (population) cross-entropy (CE) risk of a probabilistic predictor $\boldsymbol{p}(q, \Theta)$ with parameter $\Theta$,

$$\mathcal{R}_{\text{CE}}(\Theta) = \mathbb{E}[-\log p_{I^*}(\boldsymbol{Q}, \Theta)].$$

The CE loss is a *calibrated surrogate* for the 0–1 loss: its conditional minimizer predicts the true posteriors $\eta(q)$, and any sequence of models whose CE risk approaches its minimum induces decision rules whose 0–1 risk approaches the Bayes risk. Thus, training $\Theta$ by minimizing cross-entropy (with $p_i$ given by the softmax in equation 1) is statistically aligned with maximizing top-1 retrieval accuracy as shown in Proposition B.1, which follows the standard analysis of surrogate properties (Tewari & Bartlett, 2007; Bartlett et al., 2006).

**Proposition B.1.** *Let $(\boldsymbol{Q}, I^*)$ be distributed according to some unknown law. For any measurable $\boldsymbol{p}(q, \Theta) \in \Delta^{I-1}$ and the induced classifier $g_\Theta(q) := \arg\max_i p_i(q, \Theta)$, define*

$$\eta_i(q) := \Pr(I^* = i \mid \boldsymbol{Q} = q), \qquad \Delta(q) := \eta_{(1)}(q) - \eta_{(2)}(q),$$

*where $\eta_{(1)} \geq \eta_{(2)} \geq \cdots$ are the sorted coordinates of $\eta(q)$. Then:*

(i) *(Conditional optimality) For each fixed $q$, the conditional CE risk*

$$\mathcal{L}(\boldsymbol{p}; \eta(q)) := \mathbb{E}[-\log p_{I^*} \mid \boldsymbol{Q} = q] = -\sum_{i=1}^{I} \eta_i(q) \log p_i$$

*is uniquely minimized over $\boldsymbol{p} \in \Delta^{I-1}$ at $\boldsymbol{p} = \eta(q)$.*

(ii) *(Excess-risk decomposition)*

$$\mathcal{R}_{\text{CE}}(\Theta) - \inf_{\boldsymbol{p}} \mathcal{R}_{\text{CE}} = \mathbb{E}\big[\text{KL}\big(\eta(\boldsymbol{Q}) \,\|\, \boldsymbol{p}(\boldsymbol{Q}, \Theta)\big)\big],$$

*where $\inf_{\boldsymbol{p}} \mathcal{R}_{\text{CE}} = \mathbb{E}\big[H(\eta(\boldsymbol{Q}))\big]$, with $H$ the Shannon entropy.*

(iii) *(Bayes consistency / classification calibration) Suppose $\Pr\big(\Delta(\boldsymbol{Q}) = 0\big) = 0$ (no ties almost surely). If a sequence $\Theta_n$ satisfies $\mathcal{R}_{\text{CE}}(\Theta_n) \to \inf_{\boldsymbol{p}} \mathcal{R}_{\text{CE}}$, then*

$$\mathcal{R}_{0\text{--}1}\big(g_{\Theta_n}\big) \longrightarrow \inf_{g} \mathcal{R}_{0\text{--}1} = \mathcal{R}_{0\text{--}1}(g^\star).$$

*Proof.* *(i)* For fixed $\eta$, $\mathcal{L}(\boldsymbol{p}; \eta) = -\sum_i \eta_i \log p_i$ is minimized at $\boldsymbol{p} = \eta$ by Gibbs' inequality, since

$$-\sum_i \eta_i \log p_i = H(\eta) + \text{KL}(\eta \| \boldsymbol{p}) \geq H(\eta),$$

with equality iff $\boldsymbol{p} = \eta$.

*(ii)* Taking expectation over $\boldsymbol{Q}$ in the identity above yields

$$\mathcal{R}_{\text{CE}}(\Theta) = \mathbb{E}[H(\eta(\boldsymbol{Q}))] + \mathbb{E}\big[\text{KL}\big(\eta(\boldsymbol{Q}) \,\|\, \boldsymbol{p}(\boldsymbol{Q}, \Theta)\big)\big],$$

and the infimum over all $\boldsymbol{p}$ is attained by $\boldsymbol{p} = \eta$ pointwise.

*(iii)* By (ii), $\mathcal{R}_{\mathrm{CE}}(\boldsymbol{\Theta}_n) \downarrow \inf \mathcal{R}_{\mathrm{CE}}$ implies $\mathbb{E}\big[\mathrm{KL}\big(\eta(\boldsymbol{Q}) \,\|\, \boldsymbol{p}(\boldsymbol{Q}, \boldsymbol{\Theta}_n)\big)\big] \to 0$. Pinsker's inequality gives, for each $n$,

$$\|\eta(\boldsymbol{Q}) - \boldsymbol{p}(\boldsymbol{Q}, \boldsymbol{\Theta}_n)\|_{\mathrm{TV}} \;\leq\; \sqrt{\tfrac{1}{2}\,\mathrm{KL}\big(\eta(\boldsymbol{Q}) \,\|\, \boldsymbol{p}(\boldsymbol{Q}, \boldsymbol{\Theta}_n)\big)},$$

hence the total variation distance converges to 0 in $L^1$ and along a subsequence almost surely. Wherever $\Delta(\boldsymbol{Q}) > 0$, this forces $\arg\max_i p_i(\boldsymbol{Q}, \boldsymbol{\Theta}_n) = \arg\max_i \eta_i(\boldsymbol{Q})$ for all large $n$. Therefore $g_{\boldsymbol{\Theta}_n}(\boldsymbol{Q}) \to g^\star(\boldsymbol{Q})$ almost surely, and by bounded convergence, $\mathcal{R}_{0\text{-}1}(g_{\boldsymbol{\Theta}_n}) \to \mathcal{R}_{0\text{-}1}(g^\star)$. $\qquad\square$

**Remark.** In our formulation, $p_i(\boldsymbol{q}, \boldsymbol{\Theta})$ is the softmax in equation 1, which maps any score vector to a valid probability vector. Minimizing the sample average of $-\log p_{i^*}(\boldsymbol{q}, \boldsymbol{\Theta})$ is therefore an empirical proxy for minimizing $\mathcal{R}_{\mathrm{CE}}(\boldsymbol{\Theta})$, and by the proposition it targets the Bayes-optimal top-1 retrieval rule under the 0–1 criterion. We also note that the retrieval probabilities $p_i(\boldsymbol{q}, \boldsymbol{\Theta})$ are induced directly by the embedding scores and could potentially be improved via calibration (Chen & Mueller, 2023; Liu et al., 2024a;b; Nikitin et al., 2024), which is an interesting direction for future work.

## B.2 More efficient gradient update

Relative to using fixed embeddings (e.g., $\boldsymbol{\Theta}_1$), Algorithm 1 adds only two per-round operations: gradient computation and embeddings update. For very large tool catalogs, an even more efficient variant updates only the chosen item $\boldsymbol{\theta}_{t,i_t}$ each round: compute the (stochastic) gradient estimate for the sampled item $i_t$,

$$\boldsymbol{g}_{t,i_t} = \left(1 - \frac{\mathbb{1}\{i_t = i_t^*\}}{p_{t,i_t}}\right) \boldsymbol{q}_t,$$

and update

$$\boldsymbol{\theta}_{t+1,i} = \begin{cases} \boldsymbol{\theta}_{t,i} - \eta\,\boldsymbol{g}_{t,i_t}, & \text{if } i = i_t, \\ \boldsymbol{\theta}_{t,i}, & \text{otherwise.} \end{cases}$$

With a similar analysis of Lemma 3.1, we can show $\boldsymbol{g}'_{t,i} = \mathbb{1}\{i = i_t\}\left(1 - \frac{\mathbb{1}\{i_t = i_t^*\}}{p_{t,i_t}}\right)\boldsymbol{q}_t$ (which matches the above variant update by noting the indicator $\mathbb{1}\{i = i_t\}$ nulls the unchosen items) is also an unbiased estimator for gradients for all $i$. Because only the sampled item is modified at each iteration, this variant is attractive when the number of items is large.

## B.3 Variants of Algorithm 1

---

**Algorithm 2** ORAG with $K$ retrievals

---

**Input:** Initial embeddings $\boldsymbol{\Theta}_1 \in \mathbb{R}^{I \times d}$; learning rate $\eta > 0$; beam size $K \in \{1, \dots, I\}$; reranker $\mathrm{RERANK}(\boldsymbol{q}, \mathcal{I}) \to i \in \mathcal{I}$

1: **for** $t = 1, 2, \dots$ **do**
2:      Observe query embedding $\boldsymbol{q}_t \in \mathbb{R}^d$.
3:      Compute sampling probabilities from current $\boldsymbol{\Theta}_t$ via equation 1:

$$p_{t,i} = p_i(\boldsymbol{q}_t, \boldsymbol{\Theta}_t), \quad i = 1, \dots, I.$$

4:      Sample a set $\mathcal{I}_t$ of size $K$ *without replacement* from $\boldsymbol{p}_t = (p_{t,1}, \dots, p_{t,I})$.
5:      Obtain final choice $i_t \leftarrow \mathrm{RERANK}(\boldsymbol{q}_t, \mathcal{I}_t)$ and observe feedback $\mathbb{1}\{i_t = i_t^*\}$.
6:      Compute the (stochastic) gradient estimate $\boldsymbol{g}_{t,i}$ for each item $i$:

$$\boldsymbol{g}_{t,i} = \left(p_{t,i} - \frac{\mathbb{1}\{i = i_t\}\mathbb{1}\{i_t = i_t^*\}}{p_{t,i_t}}\right) \boldsymbol{q}_t. \qquad (5)$$

7:      Update embeddings for $\boldsymbol{\Theta}_{t+1} = [\boldsymbol{\theta}_{t+1,1}, \dots, \boldsymbol{\theta}_{t+1,I}]^\top$:

$$\boldsymbol{\theta}_{t+1,i} = \boldsymbol{\theta}_{t,i} - \eta \cdot \boldsymbol{g}_{t,i}.$$

8: **end for**

---

---

**Algorithm 3** ORAG with Dynamic Database

---

**Input:** Initial item set $\mathcal{I}_1$ and embeddings $\boldsymbol{\Theta}_1 \in \mathbb{R}^{|\mathcal{I}_1| \times d}$; learning rate $\eta > 0$; initializer INITEMBED$(i) \in \mathbb{R}^d$ for new items
1: **for** $t = 1, 2, \ldots$ **do**
2:      Observe current available item set $\mathcal{I}_t$ (additions/removals relative to $\mathcal{I}_{t-1}$).
3:      Maintain embeddings for $\boldsymbol{\Theta}_t$:
          • For each $i \in \mathcal{I}_t \setminus \mathcal{I}_{t-1}$ (new item), add a row $\boldsymbol{\theta}_{t,i} \leftarrow$ INITEMBED$(i)$ to $\boldsymbol{\Theta}_t$.
          • For each $i \in \mathcal{I}_{t-1} \setminus \mathcal{I}_t$ (removed item), delete row $\boldsymbol{\theta}_{t-1,i}$ from $\boldsymbol{\Theta}_{t-1}$.
4:      Observe query embedding $\boldsymbol{q}_t \in \mathbb{R}^d$.
5:      Compute probabilities over available items via equation 1:

$$p_{t,i} = p_i(\boldsymbol{q}_t, \boldsymbol{\Theta}_t), \quad i \in \mathcal{I}_t.$$

6:      Sample $i_t \sim \boldsymbol{p}_t$ and observe $\mathbb{1}\{i_t = i_t^*\}$.
7:      Compute the (stochastic) gradient estimate $\boldsymbol{g}_{t,i}$ for each item $i$:

$$\boldsymbol{g}_{t,i} = \left( p_{t,i} - \frac{\mathbb{1}\{i = i_t\}\mathbb{1}\{i_t = i_t^*\}}{p_{t,i_t}} \right) \boldsymbol{q}_t. \tag{6}$$

8:      Update embeddings for $\boldsymbol{\Theta}_{t+1} = [\boldsymbol{\theta}_{t+1,1}, \ldots, \boldsymbol{\theta}_{t+1,I}]^\top$:

$$\boldsymbol{\theta}_{t+1,i} = \boldsymbol{\theta}_{t,i} - \eta \cdot \boldsymbol{g}_{t,i}.$$

9: **end for**

---

**Algorithm 4** ORAG with Multi-Hop

---

**Input:** Initial embeddings $\boldsymbol{\Theta}_1 = [\boldsymbol{\theta}_{1,1}, \ldots, \boldsymbol{\theta}_{1,I}]^\top \in \mathbb{R}^{I \times d}$; learning rate $\eta > 0$
1: **for** $t = 1, 2, \ldots$ **do**
2:      Observe a sequence of sub-task embeddings $\mathcal{Q}_t = \{\boldsymbol{q}_t^{(h)}\}_{h=1}^{H_t}$.
3:      Initialize $\boldsymbol{\Theta}_t^{(1)} \leftarrow \boldsymbol{\Theta}_t$.
4:      **for** $h = 1, 2, \ldots, H_t$ **do**              ▷ sub-tasks within round $t$
5:          Compute sampling probabilities via equation 1:

$$p_{t,h,i} = p_i\left( \boldsymbol{q}_t^{(h)}, \boldsymbol{\Theta}_t^{(h)} \right), \quad i = 1, \ldots, I. \tag{7}$$

6:          Sample an item $i_{t,h} \sim \boldsymbol{p}_{t,h} = (p_{t,h,1}, \ldots, p_{t,h,I})$ and obtain judge feedback $y_{t,h} \in \{0, 1\}$.
7:          Compute the (stochastic) gradient estimate $\boldsymbol{g}_{t,h,i}$ for each item $i$:

$$\boldsymbol{g}_{t,h,i} = \left( p_{t,h,i} - \frac{\mathbb{1}\{i = i_{t,h}\}\, y_{t,h}}{p_{t,h,i_{t,h}}} \right) \boldsymbol{q}_t^{(h)}. \tag{8}$$

8:          Update embeddings:
$$\boldsymbol{\theta}_{t,i}^{(h+1)} \leftarrow \boldsymbol{\theta}_{t,i}^{(h)} - \eta \cdot \boldsymbol{g}_{t,h,i}.$$

9:      **end for**
10:     Set $\boldsymbol{\Theta}_{t+1} \leftarrow \boldsymbol{\Theta}_t^{(H_t+1)}$.
11: **end for**

---

## B.4 NOISY BINARY FEEDBACK

In the main text we assumed that, after sampling an item $i_t \sim p_t$, the learner observes the exact bandit feedback $\mathbf{1}\{i_t = i_t^\star\} \in \{0, 1\}$. In many deployments this binary signal can itself be noisy (e.g., due to imperfect automatic judges or inconsistent user feedback). In this section we extend the regret analysis to the case where the feedback bit is flipped with some probability. The numerical experiments are delayed to Appendix C.5.3 (Figure 11).

**Setup.** We keep the online RAG setting and the notation. At round $t$ a query embedding $q_t \in \mathbb{R}^d$ arrives, the retriever forms probabilities $p_{t,i} = p_i(q_t, \Theta_t)$ via the softmax in equation (1), samples $i_t \sim p_t$, and incurs cross-entropy loss $l(\Theta_t; (q_t, i_t^\star)) = -\log p_{t,i_t^\star}$, where $i_t^\star$ is the (unobserved) optimal item. As before, the offline optimum is

$$\Theta^\star = \arg\min_\Theta \sum_{t=1}^T l(\Theta; (q_t, i_t^\star)).$$

Instead of observing the clean indicator $r_t := \mathbf{1}\{i_t = i_t^\star\}$, we now receive a noisy binary feedback $\tilde{y}_t \in \{0, 1\}$ that is flipped with probability $\rho \in [0, \frac{1}{2})$:

$$\Pr(\tilde{y}_t = r_t \mid i_t, i_t^\star) = 1 - \rho, \qquad \Pr(\tilde{y}_t = 1 - r_t \mid i_t, i_t^\star) = \rho. \tag{9}$$

We now run Algorithm 1, except that in Step 4 the feedback $1\{i_t = i_t^\star\}$ is replaced by $\tilde{y}_t$:

$$g_{t,i}^{\text{noisy}} = \left( p_{t,i} - \frac{\mathbf{1}\{i = i_t\} \tilde{y}_t}{p_{t,i_t}} \right) q_t, \qquad i = 1, \dots, I. \tag{10}$$

We refer to this as *ORAG with noisy feedback.*

**Bias of the gradient estimator.** Let $G_t^{\text{noisy}} = [g_{t,1}^{\text{noisy}}, \dots, g_{t,I}^{\text{noisy}}]^\top$ denote the matrix of stochastic gradients in equation 10, and let

$$\tilde{G}_t = \left[ \tilde{g}_{t,1}, \dots, \tilde{g}_{t,I} \right]^\top, \qquad \tilde{g}_{t,i} = \frac{\partial}{\partial \theta_i} l(\Theta; (q_t, i_t^\star)) \Big|_{\Theta = \Theta_t} = (p_{t,i} - \mathbf{1}\{i = i_t^\star\}) q_t$$

be the full-information gradient matrix. A direct calculation shows that, conditioned on $(q_t, \Theta_t, i_t^\star)$,

$$\mathbb{E}\left[g_{t,i}^{\text{noisy}}\right] = (p_{t,i} - \mathbb{E}[\tilde{y}_t \mid i_t = i]) q_t = \begin{cases} (p_{t,i_t^\star} - (1 - \rho)) q_t, & i = i_t^\star, \\ (p_{t,i} - \rho) q_t, & i \neq i_t^\star, \end{cases} \tag{11}$$

where we used that $i_t = i_t^\star$ if and only if $r_t = 1$. Comparing this with the clean gradient $\tilde{g}_{t,i} = (p_{t,i} - \mathbf{1}\{i = i_t^\star\}) q_t$, the bias matrix $B_t := \mathbb{E}[G_t^{\text{noisy}}] - \tilde{G}_t$ has rows

$$b_{t,i} := \mathbb{E}\left[g_{t,i}^{\text{noisy}}\right] - \tilde{g}_{t,i} = \begin{cases} +\rho \, q_t, & i = i_t^\star, \\ -\rho \, q_t, & i \neq i_t^\star, \end{cases} \tag{12}$$

so that

$$\|B_t\|_F = \rho \sqrt{I} \, \|q_t\|_2. \tag{13}$$

We now quantify how this bias affects the regret.

**Theorem B.2** (Regret with noisy feedback). *Assume the noise model equation 9 with $\rho \in [0, \frac{1}{2})$ and run Algorithm 1 using the noisy update equation 10. Then, for any sequence $\{(q_t, i_t^\star)\}_{t=1}^T$,*

$$\mathbb{E}\left[\text{Reg}_{\text{ORAG}}^{noisy}\left(\{(q_t, i_t^\star)\}_{t=1}^T\right)\right] = \mathbb{E}\left[\sum_{t=1}^T l(\Theta_t; (q_t, i_t^\star)) - \sum_{t=1}^T l(\Theta^\star; (q_t, i_t^\star))\right]$$

$$\leq \frac{\|\Theta_1 - \Theta^\star\|_F^2}{2\eta} + \eta \sum_{t=1}^T \left( 1 - 2p_{t,i_t^\star} + \frac{1}{p_{t,i_t^\star}} + \rho \sum_{i=1}^I \frac{1}{p_{t,i}} \right) \|q_t\|_2^2$$

$$- \sum_{t=1}^T \mathbb{E}\left[\langle B_t, \Theta_t - \Theta^\star \rangle_F\right],$$

*where $p_{t,i} = p_i(q_t, \Theta_t)$, $B_t$ is the bias matrix defined in equation 12, and the expectation is over both $i_t \sim p_t$ and the noisy labels $\tilde{y}_t$.*

*Proof.* The proof follows the analysis of Theorem 5.1 with two additional steps to handle the bias. The online gradient descent algebra still yields

$$\sum_{t=1}^T \langle G_t^{\text{noisy}}, \Theta_t - \Theta^\star \rangle_F \leq \frac{\|\Theta_1 - \Theta^\star\|_F^2}{2\eta} + \frac{\eta}{2} \sum_{t=1}^T \|G_t^{\text{noisy}}\|_F^2.$$

Convexity of $l(\cdot;(q_t, i_t^\star))$ gives

$$l(\Theta_t;(q_t, i_t^\star)) - l(\Theta^\star;(q_t, i_t^\star)) \leq \langle \tilde{G}_t, \Theta_t - \Theta^\star \rangle_F = \langle \mathbb{E}[G_t^{\text{noisy}}] - B_t, \Theta_t - \Theta^\star \rangle_F.$$

Taking expectations and summing over $t$ yields

$$\mathbb{E}[\text{Reg}_{\text{ORAG}}^{\text{noisy}}] \leq \frac{\|\Theta_1 - \Theta^\star\|_F^2}{2\eta} + \frac{\eta}{2}\sum_{t=1}^{T}\mathbb{E}\big[\|G_t^{\text{noisy}}\|_F^2\big] - \sum_{t=1}^{T}\mathbb{E}\big[\langle B_t, \Theta_t - \Theta^\star \rangle_F\big].$$

For the second-moment term, write $G_t^{\text{noisy}} = G_t + \Delta_t$, where $G_t$ is the unbiased estimator using the clean indicator $r_t$ (as in Section 3) and $\Delta_t$ is the perturbation induced by noise: $\Delta_{t,i} = -\mathbf{1}\{i = i_t\}(\tilde{y}_t - r_t)q_t/p_{t,i_t}$. Then $\|G_t^{\text{noisy}}\|_F^2 \leq 2\|G_t\|_F^2 + 2\|\Delta_t\|_F^2$, and the bound from Theorem 5.1 gives $\mathbb{E}[\|G_t\|_F^2] \leq (1 - 2p_{t,i_t^\star} + 1/p_{t,i_t^\star})\|q_t\|_2^2$. Moreover, $\|\Delta_t\|_F^2 = (\tilde{y}_t - r_t)^2\|q_t\|_2^2/p_{t,i_t}^2$, and since $(\tilde{y}_t - r_t)^2 = 1$ with probability $\rho$ and $0$ otherwise, we obtain

$$\mathbb{E}\big[\|\Delta_t\|_F^2\big] = \rho\sum_{i=1}^{I}\frac{1}{p_{t,i}}\|q_t\|_2^2.$$

Combining these inequalities yields the claimed second line of the bound. $\square$

The last term $\sum_t \mathbb{E}[\langle B_t, \Theta_t - \Theta^\star\rangle_F]$ captures the effect of the bias. Its sign is not fixed in general, but it can be upper-bounded in terms of the noise level and the diameter of the parameter set.

**Corollary B.3** (Explicit noise dependence under bounded embeddings). *Assume the conditions of Theorem B.2 and, in addition, that*

- *each item embedding is projected onto a ball $\{\theta \in \mathbb{R}^d : \|\theta\|_2 \leq R\}$ after every update (as suggested in Algorithm 1), so that $\|\Theta_t - \Theta^\star\|_F \leq 2R\sqrt{I}$ for all $t$;*

- *there exist constants $\bar{\Theta} > 0$, $p \in (0,1)$, $\underline{p} \in (0, 1/I)$ and $\bar{q} > 0$ such that $\|\Theta_1 - \Theta^\star\|_F^2 \leq \bar{\Theta}$, $p_{t,i_t^\star} \geq p$, $p_{t,i} \geq \underline{p}$ for all $i, t$, and $\|q_t\|_2^2 \leq \bar{q}$ for all $t$.*

*Then, for any $\eta > 0$,*

$$\mathbb{E}\Big[\text{Reg}_{\text{ORAG}}^{noisy}\big(\{(q_t, i_t^\star)\}_{t=1}^T\big)\Big] \leq \frac{\bar{\Theta}}{2\eta} + \eta\,\bar{q}\left(1 - 2p + \frac{1}{p} + \rho\frac{I}{\underline{p}}\right)T + 2\rho RI\sqrt{\bar{q}}\,T.$$

*In particular, by choosing*

$$\eta = \sqrt{\frac{\bar{\Theta}}{2\bar{q}\big(1 - 2p + \frac{1}{p} + \rho\frac{I}{\underline{p}}\big)T}},$$

*the regret satisfies*

$$\mathbb{E}\Big[\text{Reg}_{\text{ORAG}}^{noisy}\big(\{(q_t, i_t^\star)\}_{t=1}^T\big)\Big] = O\big(\sqrt{T}\big) + O(\rho T).$$

*Proof.* From equation 13 and the projection assumption,

$$\big|\langle B_t, \Theta_t - \Theta^\star\rangle_F\big| \leq \|B_t\|_F\,\|\Theta_t - \Theta^\star\|_F \leq \rho\sqrt{I}\,\|q_t\|_2 \cdot 2R\sqrt{I} \leq 2\rho RI\sqrt{\bar{q}},$$

so the bias contribution is at most $2\rho RI\sqrt{\bar{q}}\,T$. The remaining terms are bounded using $p_{t,i_t^\star} \geq p$, $p_{t,i} \geq \underline{p}$ and $\|q_t\|_2^2 \leq \bar{q}$ exactly as in Corollary 5.2, yielding the stated inequality and rate. $\square$

Corollary B.3 shows that with noise feedback, Algorithm 1's regret bound decomposes into a standard $O(\sqrt{T})$ term (as in the noiseless case) plus an additive $O(\rho T)$ term coming from the persistent gradient bias.

## C EXPERIMENT DETAILS

### C.1 DATASET AND PROMPT DETAILS

This part provides details on the construction of queries and document/tool representations for each benchmark. To ensure reproducibility, we outline the exact data fields and templates used to generate the text for embedding.

#### C.1.1 ULTRATOOL

Following prior work (Braunschweiler et al., 2025), we decompose each annotated plan step into a standalone retrieval query. For each sample, we construct the query from the top-level question (`question` column) and the specific plan step (`step` column) using the following template:

```
Given the following task:"{question}", select the best tool provided
in the context to solve the following substep:"{step}".
```

The resulting text is used as the input for the query embedding model.

For each tool, we create a single text representation (stored as `text_representation` column) by concatenating the following fields in order. Fields that are empty are omitted.

- **Name:** `name`
- **Description:** `description`
- **Arguments:** `arguments` (parsed as a JSON string)
- **Results:** `results` (parsed as a JSON string)

This concatenated string is used to embed the tool documentation.

#### C.1.2 TOOLRET

We use all 35 sub-tasks from the ToolRet benchmark (Shi et al., 2025). Following the original paper, we use an instruction-based format for queries, concatenating the provided `instruction` and the user `query`:

```
{instruction}\n{query}
```

For tool documentation, we perform a schema-aware extraction from the raw JSON object. We extract and join the following fields with newlines:

- **ToolRet-Code:** `name`, `description`, `func_description`, `functionality`
- **ToolRet-Web/Customized:** `name`, `description`

If a field is not present or the documentation is not a valid JSON object, we fall back to using the raw documentation string for embedding.

#### C.1.3 FIQA

For the FiQA benchmark (Thakur et al., 2021), we follow the standard setup from the MTEB toolkit (Muennighoff et al., 2022). For both corpus and queries, we embed the content of the `text` field.

#### C.1.4 MULTIHOPRAG

For each original `query`, we generate a sequence of sub-queries (`decomposed_questions`) using an LLM-based query decomposition strategy. We use the following prompt for this task:

> You are an expert research analyst specializing in breaking down complex questions into a logical sequence of simple, answerable sub-questions.
> Your task is to decompose a given 'Original Question' into a series of smaller, ordered sub-questions. This decomposition will be used to query a retrieval system containing various factual reports.
> **Your Goal:** Create a step-by-step reasoning path. The answer to a later sub-question should ideally depend on or build upon the answer to a previous one, creating a logical chain.
> **Key Constraints:**
> 1. **Logical Flow:** The sub-questions must follow a logical order. The sequence should represent the steps a human researcher would take to find the final answer.
> 2. **Self-Contained:** Each sub-question must be understandable and answerable on its own.
> 3. **Fact-Focused:** All sub-questions must be aimed at retrieving factual information from the reports. Do not ask about the publication source or publisher unless it is essential for resolving ambiguity.
> 4. **Completeness:** The combined answers to your sub-questions should contain all the information necessary to answer the Original Question.
> 5. **No Direct Answers:** Do not try to answer the Original Question yourself. Only generate the sub-questions.

We then form a compact retrieval string (`formatted_query`) for each sub-question using the template:

```
Context: {original_query} | Focus: {sub_question}
```

For the document corpus, we create a standardized text representation by sorting all key-value pairs of a document's JSON object and joining them into a single string with the format `{key}:{value}` on each line. This approach ensures a consistent representation that includes all available information (e.g., category, title, body).

### C.1.5 COMMON PREPROCESSING AND EMBEDDING DETAILS

Before embedding, we apply light text normalization to all inputs, including stripping whitespace and replacing newlines for API stability. If an input exceeds the model's length limit, we progressively truncate it (e.g., to 8192 characters and then shorter) and skip any samples that remain too long. The output dimension for all embedding models is set to 1536.

### C.2 TRAINER

Our algorithm is implemented using PyTorch. We employ the AdamW optimizer with default parameters and a learning rate schedule that decays proportionally to $1/\sqrt{t}$, where $t$ is the training/update step. Key hyperparameters, including the initial learning rate and batch size, were tuned via a Bayesian-optimization-based grid search on the validation data (10% of the total dataset size). The search space for each hyperparameter is detailed below:

- **Initial Learning Rate ($\eta_0$):** $\{$1e-8, 2e-8, 5e-8, 1e-7, 2e-7, 5e-7, 1e-6, 2e-6, 5e-6, 1e-5$\}$
- **Batch Size:** $\{5, 10, 20, 30, 40, 50\}$

### C.3 DATA ENHANCEMENT

The main results presented in Section 4.1 utilize a data augmentation strategy where each query is processed multiple times to accelerate convergence. We refer to this as the multiple exposure setting.

For a more realistic online deployment scenario, we also evaluate a single exposure setting where each query is seen only once. Table 2 presents the results for this setting. For this experiment, we used the same hyperparameters tuned for the multiple exposure setting. We note that performance could likely be further improved by re-tuning the hyperparameters specifically for the single exposure scenario.

Table 2: Retrieval performance in the single exposure setting (no data augmentation). Our method still consistently improves over the base dense retrieval models, albeit with smaller margins than in the multiple exposure setting reported in the main paper.

| Method | U-Tool | | FiQA | | T-Web | | T-Code | | T-Custom | |
|---|---|---|---|---|---|---|---|---|---|---|
| | R@10 | N@10 | R@10 | N@10 | R@10 | N@10 | R@10 | N@10 | R@10 | N@10 |
| text-embedding-v4 | 0.7451 | 0.5064 | 0.5335 | 0.4604 | 0.2701 | 0.1453 | 0.5291 | 0.3770 | 0.5066 | 0.4097 |
| text-embedding-3-large | 0.8356 | 0.6067 | 0.6258 | 0.5462 | 0.3243 | 0.1675 | 0.5347 | 0.3582 | 0.6378 | 0.5204 |
| **Ours (text-emb.-v4)** | $0.7614_{\pm.011}$ (+1.63%) | $0.5209_{\pm.010}$ (+1.45%) | $0.5382_{\pm.013}$ (+0.47%) | $0.4646_{\pm.009}$ (+0.42%) | $0.2900_{\pm.023}$ (+1.99%) | $0.1579_{\pm.015}$ (+1.26%) | $0.5484_{\pm.017}$ (+1.93%) | $0.3891_{\pm.014}$ (+1.21%) | $0.5202_{\pm.011}$ (+1.36%) | $0.4237_{\pm.012}$ (+1.40%) |
| **Ours (text-emb.-3-L.)** | $0.8540_{\pm.014}$ (+1.86%) | $0.6180_{\pm.012}$ (+1.13%) | $0.6284_{\pm.015}$ (+0.26%) | $0.5483_{\pm.012}$ (+0.21%) | $0.3458_{\pm.021}$ (+2.15%) | $0.1804_{\pm.019}$ (+1.29%) | $0.5430_{\pm.020}$ (+0.83%) | $0.3671_{\pm.015}$ (+0.89%) | $0.6561_{\pm.014}$ (+1.83%) | $0.5324_{\pm.009}$ (+1.20%) |

## C.4 MORE DETAILS ON THE EXPERIMENTS

We offer more details of the experiments included in our main paper here.

**Illustration experiments.** The experiment illustrated in Figure 1a is conducted on a subset of the *ToolRet-Code* dataset. We only use the tool items whose `documentation` column contains a `functionality` key, where the corresponding value is a highly compact and ambiguous description of the tool item. We refer to the full documentation as a "good documentation", and the only `functionality` description as a "bad documentation". The visualization in Figure 1b is extracted from an experiment run on *UltraTool* dataset in Table 1. We randomly sample 100 tool items and inspect their embeddings and performance metrics across different settings.

**Variant experiments.** All the variant experiments are conducted without the data enhancement techniques mentioned in Section C.3 to evaluate the practical performance under an online setting. Also, considering the high costs of LLM-involved experiments, we did not fully tune the parameters during experiments integrated with the LLM reranker, and only ran 1 round of them.

**Varying database experiment.** For the varying database experiment shown in Figure 5, we provide half of the tools in the beginning, and only add the other half as available tools when half of the queries are processed. The queries are not manipulated.

**Multi-Hop retrieval experiment.** Figure 7 provides a visual depiction of the combined online-optimizing and inference workflow for the multi-hop retrieval experiments shown in Figure 6. The multi-hop pipeline uses an LLM agent (backed by `gpt-4o-mini-2024-07-18`) for two key steps: (1) reranking retrieved documents for each sub-task query, and (2) synthesizing a final answer from the collected evidence, with the support of OpenAI's JSON mode. The prompts for these steps are provided below.

> You are an impartial and meticulous AI judge. Your task is to determine which of the provided documents contains useful information to answer the given question, especially the "Focus" one.
> Carefully review each document and respond with a JSON object containing the 0-based index of the relevant document. A smaller index is more relevant.
> Question:
> {question}
> Retrieved Documents:
> {formatted_docs}
> Based on the question, which document is the most relevant?

> You are a concise QA assistant. Given a main question and evidence documents, provide the final short answer only. If uncertain, provide your best effort.
> Main Question:
> {question}
> Evidence Documents:
> {formatted_joined_docs}
> Provide the final answer only with no explanation.

**Regret analysis experiment.** Figure 8 empirically verifies a sublinear cumulative regret for Algorithm 1. We perform our regret analysis on the *ToolRet-Code* dataset, where the result is displayed in Figure 8. To evaluate performance across different time horizons ($T$), we truncate the query set

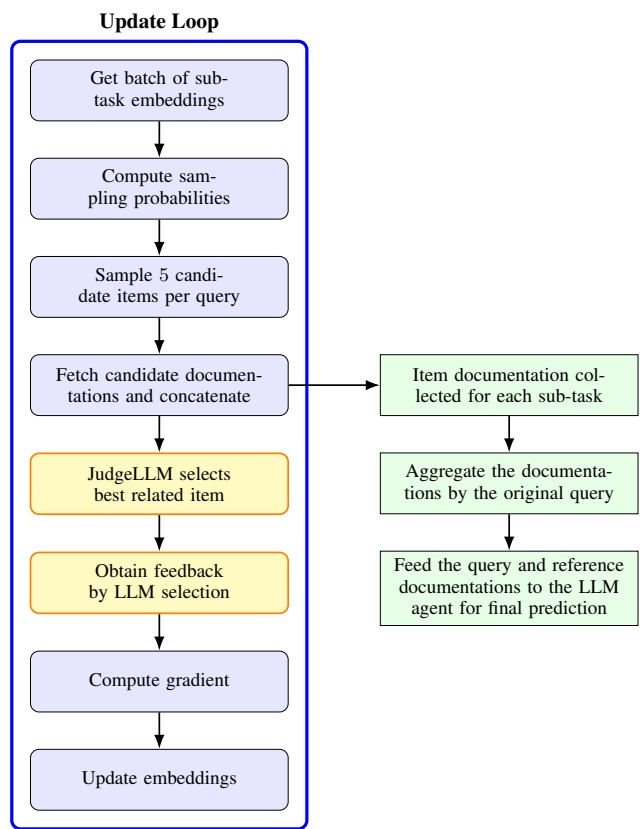

Figure 7: Workflow for the multi-hop experiment. The left panel shows the update loop for our Algorithm 4, which leverages the decomposed sub-task query embeddings and optimizes the document embedding. The right panel illustrates the inference process where, for each sub-query, retrieved documents are collected and then synthesized by an LLM agent to produce the final answer.

to various lengths while keeping all other hyperparameters identical to those used for the results in Table 1. Regret is calculated as the difference between the cumulative loss of our online Algorithm 1 and an oracle baseline trained with full-information gradients (see Section 3), with the cross-entropy loss being the loss function.

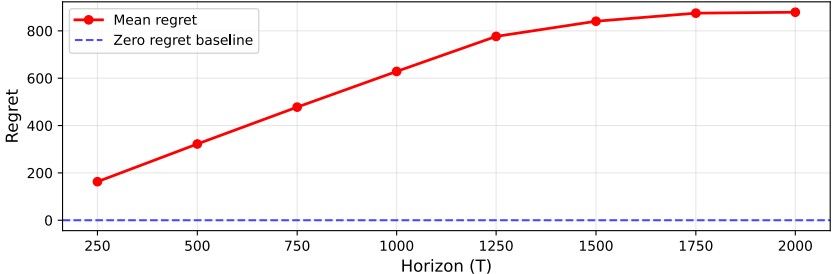

Figure 8: Cumulative regret for Algorithm 1.

## C.5 ADDITIONAL EXPERIMENTS

In this section, we conduct additional experiments under broader practical settings.

### C.5.1 Temporal Performance Evaluation

We analyze the temporal dynamics of the model's performance throughout the optimization process. Specifically, we employ two distinct evaluation protocols:

1. **Offline Evaluation (Classic Setting):** The dataset is partitioned into a training set ($80\%$) and a hold-out evaluation set ($20\%$). The algorithm updates its embeddings using only the training set, and performance is reported on the static evaluation set.

2. **Online Evaluation (Deployment Setting):** We simulate a continuous deployment scenario where we track performance on queries as they are processed. We report metrics on *seen queries*, where we compute the Recall@10 and NDCG@10 for each seen sample and obtain their average iteratively to monitor how the algorithm adapts to the data distribution it has encountered. Additionally, we track the metrics specifically for the *first 10 queries* encountered during the process to inspect for potential catastrophic forgetting or drift over long-term evolution.

We apply these protocols to the *UltraTool* dataset. The results are illustrated in Figure 9. At the population level, both Recall and NDCG metrics demonstrate a sustained improvement as training progresses, corroborating the results presented in Tables 1 and 2. Regarding the specific inspection of the first 10 seen queries, we observe that while the Recall metric exhibits notable variance across different runs, the NDCG metric shows a step-wise increase. Crucially, there is no sign of performance degradation over time, confirming that our method does not suffer from catastrophic forgetting.

### C.5.2 Sensitivity Analysis

We conduct a sensitivity analysis using the *UltraTool* dataset to assess the robustness of our method to hyperparameter variations and environmental noise. Unless stated otherwise, we utilize a default learning rate of $2 \times 10^{-8}$ and a batch size of $40$.

First, we investigate the impact of different learning rate schedules on final performance. As illustrated in Figure 10, we compare an inverse-square-root decay schedule, a cosine decay schedule, and a constant learning rate. The results indicate that, at similar initial learning rates, the cosine decay schedule yields performance comparable to the inverse-square-root decay. In contrast, the constant schedule consistently underperforms across all tested learning rates.

Second, we examine the system's stability with respect to batch size and feedback noise, as shown in Figure 11. To simulate noisy feedback, each binary feedback label $y_t$ is flipped with probability $\rho$ (the noise rate), such that $y_{noisy} \sim \mathrm{Bernoulli}(\rho)$ when the true label is 0, and vice versa. The results demonstrate that our method is robust to variations in both batch size and noise levels. Under moderate conditions, the system exhibits resilience, maintaining stable performance even as these factors fluctuate.

### C.5.3 More Baselines

To strengthen our empirical evaluation, we expand our comparison to include an integrated Retriever-Reranker pipeline and three representative fine-tuning baselines: DPR, ReFit, and NUDGE. All experiments are implemented on the *UltraTool* dataset.

**Retriever + Reranker Pipeline.** While the main paper analyzes the retriever and reranker components separately and only tries a chat LLM for reranking, here we evaluate a practical cascaded pipeline. In this setup, the retriever (using `text-embedding-large-3`) first selects the top-10 candidate documents. A cross-encoder-based reranker (`Qwen3-Reranker-0.6B`) then scores these candidates to determine the final selection. The results, shown in Figure 12, demonstrate that the inclusion of a reranker not only elevates baseline retrieval accuracy but also enriches the feedback signal for our online optimization. By providing a denser positive signal, the reranker further boosts the performance of our method compared to the vanilla retriever.

**Fine-Tuning Baselines.** We benchmark against three model-agnostic techniques that enhance retrieval from different perspectives:

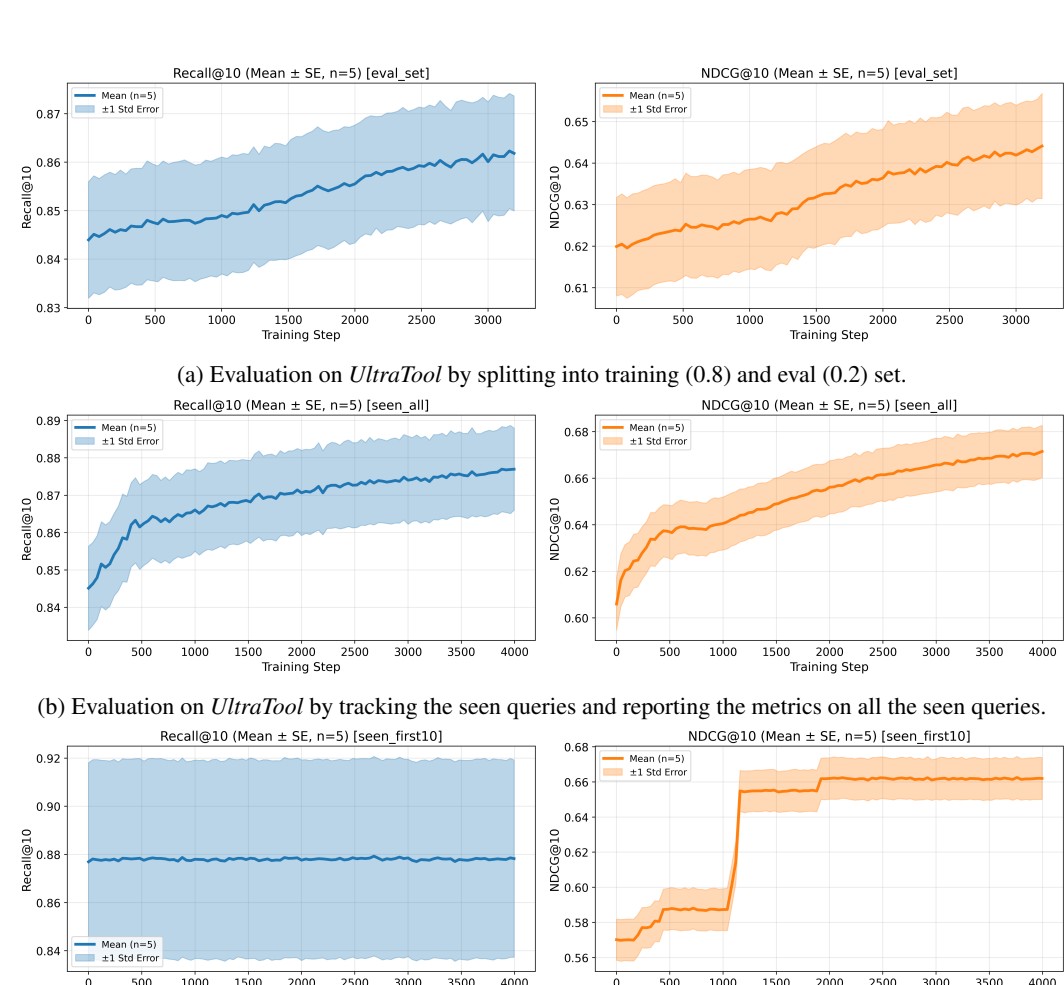

(a) Evaluation on *UltraTool* by splitting into training (0.8) and eval (0.2) set.

(b) Evaluation on *UltraTool* by tracking the seen queries and reporting the metrics on all the seen queries.

(c) Evaluation on *UltraTool* by tracking the seen queries and reporting the metrics on the first 10 seen queries.

Figure 9: Temporal performance evaluation on *UltraTool*.

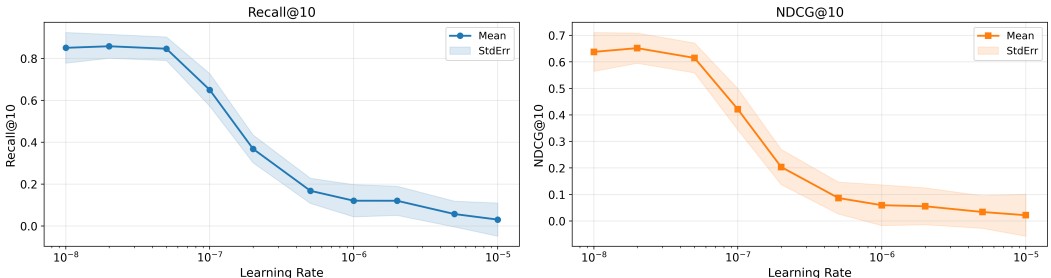

(a) Performance variation with inverse-square-root decay schedule. The learning rate decays at a rate of $1/\sqrt{t}$.

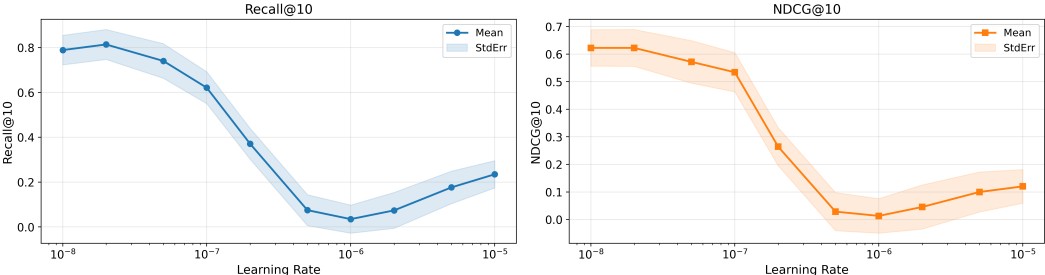

(b) Performance variation with cosine decay schedule. The learning rate decays slowly at the beginning and end of training, with a rapid decrease during the intermediate phase.

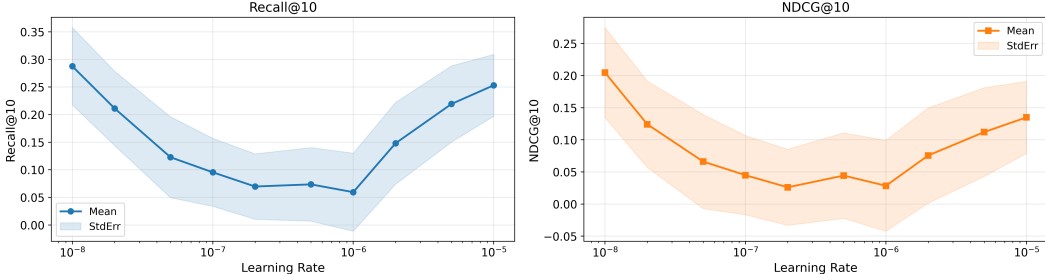

(c) Performance variation with constant schedule. The learning rate remains fixed throughout training.

Figure 10: Performance dynamics with varying batch sizes and noise rates.

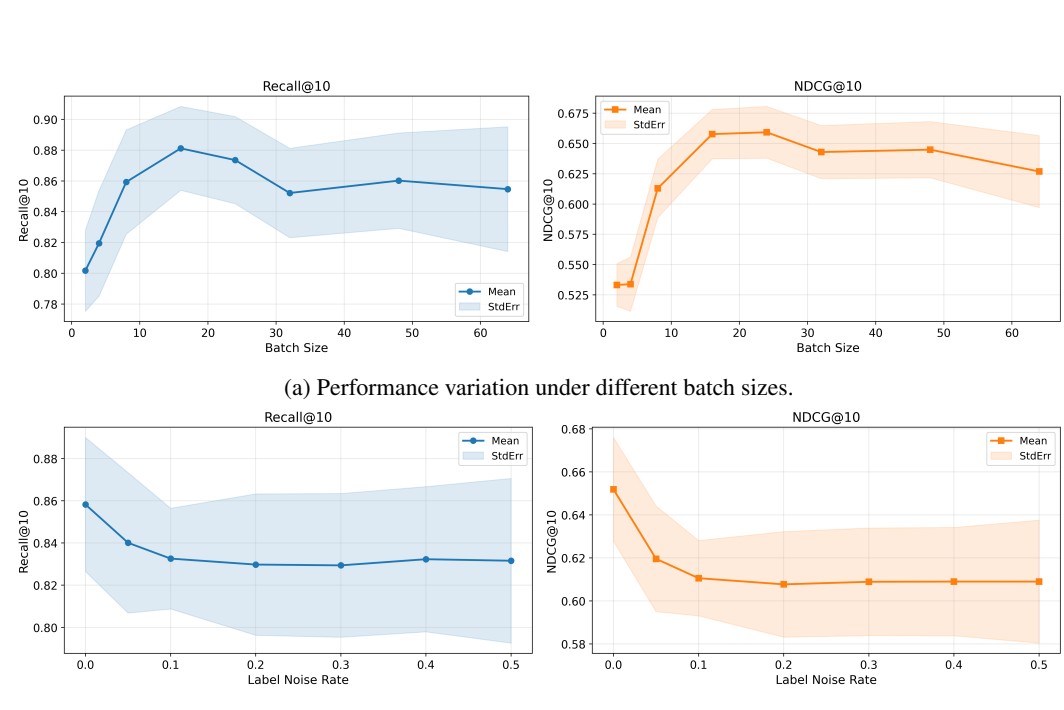

(a) Performance variation under different batch sizes.

(b) Performance variation under different label noise levels.

Figure 11: Performance dynamics with different levels of batch size and noise rate.

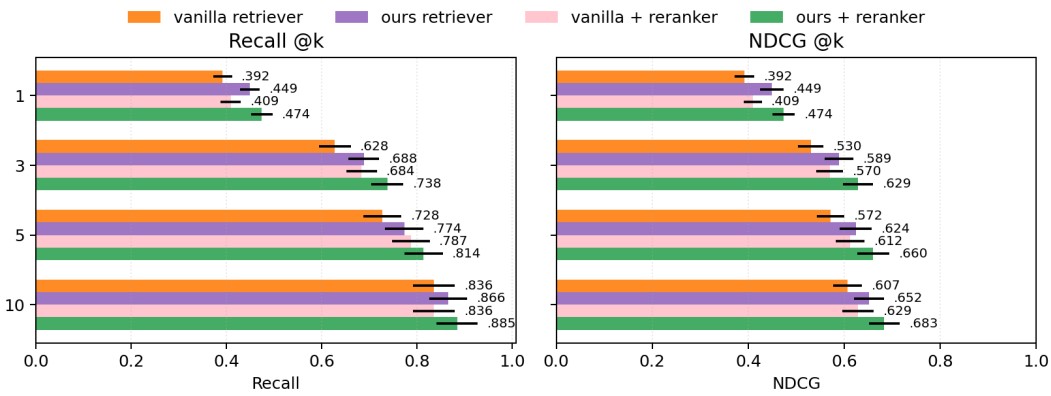

Figure 12: Comparison between the vanilla retriever and our method, with and without a reranker.

- **DPR** (Karpukhin et al., 2020): Fine-tunes the embedding model itself to improve retrieval quality.

- **ReFit** (Reddy et al., 2023): Optimizes the query-side embeddings using an offline method.

- **NUDGE** (Zeighami et al., 2024): Implements an offline optimization strategy to refine document embeddings.

Comparisons against these methods highlight the distinct advantages of our online, lightweight framework. As detailed in Table 3, while NUDGE leads in NDCG—consistent with the main strengths reported in its original publication—our method achieves the highest Recall@10 rate. This metric is particularly critical for real-world tool-use applications, where the retrieval stage typically feeds a candidate set to an LLM agent for final selection. High Recall ensures the correct tool is available within this set, directly supporting the agent's decision-making process.

Table 3: Comparison with additional baselines on the *UltraTool* dataset. Metrics reported are Recall@10 (R@10) and NDCG@10 (N@10). Standard errors are given after each score value.

| Method | R@10 | N@10 |
|---|---|---|
| text-embedding-3-large | 0.8356 | 0.6067 |
| DPR | $0.8341_{\pm.014}$ | $0.6679_{\pm.012}$ |
| Refit | $0.8420_{\pm.021}$ | $0.6194_{\pm.017}$ |
| Nudge | $0.8487_{\pm.012}$ | $0.7673_{\pm.019}$ |
| **Ours** | $0.8682_{\pm.018}$ | $0.6522_{\pm.013}$ |

# D  APPENDIX FOR PROOFS

## D.1  PROOF FOR LEMMA 3.1

*Proof.* The (full-information with $i_t^*$ observable) gradient of $l(\boldsymbol{\Theta}; (\boldsymbol{q}_t, i_t^*))$ with respect to $\boldsymbol{\theta}_i$ is

$$\left. \frac{\partial l(\boldsymbol{\Theta}; (\boldsymbol{q}_t, i_t^*))}{\partial \boldsymbol{\theta}_i} \right|_{\boldsymbol{\Theta}=\boldsymbol{\Theta}_t} = \left( p_{t,i} - \mathbb{1}\{i = i_t^*\} \right) \boldsymbol{q}_t,$$

where $p_{t,i}$ is defined in equation 3.

By the definitions, for any $i$ we have

$$\mathbb{E}_{i_t}\left[ \boldsymbol{g}_{t,i} \right] = \sum_{i_t=1}^{I} p_{t,i_t} \cdot \left( p_{t,i} - \mathbb{1}\{i = i_t\} \frac{\mathbb{1}\{i_t = i_t^*\}}{p_{t,i_t}} \right) \boldsymbol{q}_t$$

$$= p_{t,i} \cdot \left( 1 - \frac{\mathbb{1}\{i = i_t^*\}}{p_{t,i}} \right) \boldsymbol{q}_t$$

$$= \left( p_{t,i} - \mathbb{1}\{i = i_t^*\} \right) \boldsymbol{q}_t$$

$\square$

## D.2  PROOF OF THEOREM 5.1

*Proof.* The proof follows a standard regret analysis for online convex optimization. Let $\langle \cdot, \cdot \rangle_F$ denote the Frobenius inner product and $\| \cdot \|_F$ the Frobenius norm. Define the gradient (estimator) matrices $\boldsymbol{G}_t = [\boldsymbol{g}_{t,1}, \ldots, \boldsymbol{g}_{t,I}]^\top$ and $\tilde{\boldsymbol{G}}_t = [\tilde{\boldsymbol{g}}_{t,1}, \ldots, \tilde{\boldsymbol{g}}_{t,I}]^\top$, where $\boldsymbol{g}_{t,i}$ is as in Section 3 and

$$\tilde{\boldsymbol{g}}_{t,i} = \left. \frac{\partial l(\boldsymbol{\Theta}; (\boldsymbol{q}_t, i_t^*))}{\partial \boldsymbol{\theta}_i} \right|_{\boldsymbol{\Theta}=\boldsymbol{\Theta}_t}$$

is the (full-information with $i_t^*$ observable) gradient of $l(\boldsymbol{\Theta}; (\boldsymbol{q}_t, i_t^*))$ with respect to $\boldsymbol{\theta}_i$.

By the update in Algorithm 1, for each $t$,

$$\|\mathbf{\Theta}_{t+1} - \mathbf{\Theta}^*\|_F^2 = \|\mathbf{\Theta}_t - \eta \mathbf{G}_t - \mathbf{\Theta}^*\|_F^2$$
$$= \|\mathbf{\Theta}_t - \mathbf{\Theta}^*\|_F^2 - 2\eta \langle \mathbf{G}_t, \mathbf{\Theta}_t - \mathbf{\Theta}^* \rangle_F + \eta^2 \|\mathbf{G}_t\|_F^2.$$

Summing and rearranging yields

$$\sum_{t=1}^{T} \langle \mathbf{G}_t, \mathbf{\Theta}_t - \mathbf{\Theta}^* \rangle_F = \sum_{t=1}^{T} \frac{\|\mathbf{\Theta}_t - \mathbf{\Theta}^*\|_F^2 - \|\mathbf{\Theta}_{t+1} - \mathbf{\Theta}^*\|_F^2}{2\eta} + \frac{\eta}{2} \sum_{t=1}^{T} \|\mathbf{G}_t\|_F^2$$

$$= \frac{\|\mathbf{\Theta}_1 - \mathbf{\Theta}^*\|_F^2 - \|\mathbf{\Theta}_{T+1} - \mathbf{\Theta}^*\|_F^2}{2\eta} + \frac{\eta}{2} \sum_{t=1}^{T} \|\mathbf{G}_t\|_F^2$$

$$\leq \frac{\|\mathbf{\Theta}_1 - \mathbf{\Theta}^*\|_F^2}{2\eta} + \frac{\eta}{2} \sum_{t=1}^{T} \|\mathbf{G}_t\|_F^2.$$

Because $l(\mathbf{\Theta}; (\mathbf{q}_t, i_t^*))$ is convex in $\mathbf{\Theta}$, for any $t$ (conditioning on $\mathbf{q}_t, i_t^*, \mathbf{\Theta}_t$),

$$l(\mathbf{\Theta}_t; (\mathbf{q}_t, i_t^*)) - l(\mathbf{\Theta}^*; (\mathbf{q}_t, i_t^*)) \leq \langle \tilde{\mathbf{G}}_t, \mathbf{\Theta}_t - \mathbf{\Theta}^* \rangle_F = \langle \mathbb{E}[\mathbf{G}_t], \mathbf{\Theta}_t - \mathbf{\Theta}^* \rangle_F,$$

where the equality uses the unbiasedness $\mathbb{E}[\mathbf{G}_t] = \tilde{\mathbf{G}}_t$ from Lemma 3.1 (the expectation is over $i_t \sim \mathbf{p}_t$ given the history). Summing over $t$ and applying the previous bound gives

$$\sum_{t=1}^{T} \mathbb{E}\Big[ l(\mathbf{\Theta}_t; (\mathbf{q}_t, i_t^*)) - l(\mathbf{\Theta}^*; (\mathbf{q}_t, i_t^*)) \Big] \leq \frac{\|\mathbf{\Theta}_1 - \mathbf{\Theta}^*\|_F^2}{2\eta} + \frac{\eta}{2} \sum_{t=1}^{T} \mathbb{E}\big[\|\mathbf{G}_t\|_F^2\big].$$

It remains to bound $\mathbb{E}\big[\|\mathbf{G}_t\|_F^2\big]$. By definition, $\|\mathbf{G}_t\|_F^2 = \sum_{i=1}^{I} \|\mathbf{g}_{t,i}\|_2^2$, and

$$\mathbb{E}\big[\|\mathbf{G}_t\|_F^2\big] = \sum_{i=1}^{I} \mathbb{E}\big[\|\mathbf{g}_{t,i}\|_2^2\big]$$

$$= \|\mathbf{q}_t\|_2^2 \sum_{i=1}^{I} \mathbb{E}_{i_t}\left[ \left( p_{t,i} - \frac{\mathbb{1}\{i = i_t\}\mathbb{1}\{i_t = i_t^*\}}{p_{t,i_t}} \right)^2 \right]$$

$$= \|\mathbf{q}_t\|_2^2 \sum_{i=1}^{I} p_{t,i} \left( p_{t,i}^2 - 2\mathbb{1}\{i = i_t^*\} + \frac{\mathbb{1}\{i = i_t^*\}}{p_{t,i}^2} \right)$$

$$\leq \left( \frac{1}{p_{t,i_t^*}} - 2p_{t,i_t^*} + 1 \right) \|\mathbf{q}_t\|_2^2.$$

Plugging this into the previous inequality establishes the claimed bound. $\square$

### D.3   PROOF OF COROLLARY 5.2

*Proof.* Under the assumptions given with

$$\eta = \sqrt{\frac{\underline{p}\,\bar{\Theta}}{\bar{q}\,(1 - \underline{p})(1 + 2\underline{p})\,T}},$$

the corollary is a direct result of Theorem 5.1. $\square$

## E   THE USE OF LARGE LANGUAGE MODELS

We disclose our use of the Large Language Model (LLM) assistant in the preparation of this work. Its application was limited to the following support roles:

- **Manuscript Preparation:** The LLM was used for copy-editing tasks, including correcting grammatical errors and improving sentence clarity. The authors wrote all scientific claims, analyses, and the core narrative.

- **Software Development:** The LLM was used as a coding assistant to (a) refactor and improve the efficiency of human-authored code, and (b) adapt existing code from a completed experiment to function with different datasets.

All LLM-generated suggestions for both text and code were critically reviewed, edited, and approved by the authors to ensure their correctness and alignment with the paper's contributions.

