# OpenReview forum: "Online-Optimized RAG for Tool Use and Function Calling"
_ICLR.cc/2026/Conference — Submitted to ICLR 2026_

### Official Review · Reviewer_H1oD · 2025-10-27

**Soundness:** 2
**Presentation:** 2
**Contribution:** 2
**Rating:** 4
**Confidence:** 4

**Summary:**

This paper introduces Online-Optimized RAG (ORAG), a deployment-time framework to correct "embedding misalignment" in RAG systems used for tool and function calling. The authors identify that errors at retrieval time, often caused by noisy documentation or suboptimal embedding models, lead to incorrect retrieval and task failure. ORAG addresses this by continuously adapting the tool embeddings—not the LLM or query model—using minimal, real-time feedback (e.g., task success/failure) from live user interactions. This "plug-and-play" method applies lightweight online gradient updates with negligible latency.

The update mechanism is intuitive.When a selected tool succeeds, its embedding is moved toward the query embedding to increase similarity; when it fails, it is moved away. Embeddings of unselected tools are also pushed away to refine the embedding space. The authors provide a theoretical regret analysis and show empirically that ORAG consistently improves tool selection accuracy and task success across diverse benchmarks like UltraTool and ToolRet. The method is also shown to be versatile, successfully integrating with multi-hop retrieval, dynamic tool databases, and rerankers.

**Strengths:**

1. The paper offers a robust theoretical analysis to ground the proposed online learning mechanism, while the empirical results consistently demonstrate its effectiveness in improving retrieval accuracy and end-to-end task success over multiple base embedding models.

2. The authors provide a comprehensive evaluation across multiple realistic use cases, including integration with K-retrieval rerankers, handling time-varying dynamic tool databases, and applying the method to multi-hop reasoning. This broad experimentation, spanning several datasets, strongly validates the "plug-and-play" and flexible nature of the ORAG framework for real-world deployment.

**Weaknesses:**

1. Limited Comparison to Fine-Tuning Baselines: While ORAG is an online learning algorithm designed to fine-tune embeddings, the empirical evaluation primarily compares its performance against static base embedding models. The absence of a direct comparison against established or contemporary online/offline fine-tuning algorithms (e.g., in-batch negative sampling on the tool descriptions or approaches like DynamicRAG/FLAIR/NUDGE) makes it difficult to quantify the performance benefit and computational advantages of the proposed approach over existing state-of-the-art methods in the literature.

2. Unclear Differentiation from Related Work: The paper does not sufficiently clarify the benefits of ORAG against existing related work, such as DynamicRAG, FLAIR, or NUDGE, outside of the Appendix. While the method's novelty lies in its deployment-time, bandit-style updates to tool embeddings, a more explicit discussion in the main body is needed to highlight the specific practical and theoretical advantages this approach offers compared to these previous efforts, particularly those involving dynamic retrieval or online adaptation.

3. Sensitivity to Feedback: The effectiveness of ORAG relies entirely on the quality and timeliness of the bandit feedback (success/failure). The paper would benefit from a deeper analysis of the impact of noisy or delayed feedback on the stability and convergence rate of the online updates.

**Questions:**

N/A

---

> ### Author Response · Authors · 2025-11-20
>
> We thank the reviewer for taking the time to read our paper and for all the comments. We address your questions and concerns as below:
>
> - **Comparison with Other Online Methods**: We want to first clarify that our algorithm is a lightweight deployment-time adaptation of general fully trained base retrievers to improve their performance. It is plug-and-play and can be applied to any retriever. Thus, for any off-line fine-tuning methods can be seamlessly combined with our approach - they are not competitors. The current online methods, such as AutoRAG-HP, MBA-RAG, and DynamicRAG, adapt controller-level hyperparameters (e.g., when to retrieve, top-k, or reranking policies) using bandit or RL techniques, but they treat the underlying retriever system as fixed. ORAG instead adapts the retrieval geometry itself by updating item embeddings online. These axes of adaptation are also complementary: (i) Controller methods decide how much or when to retrieve; (ii) ORAG decides how queries map to items in the embedding space. In summary, our existing experiments are designed to highlight the improvements over base retrievers after applying our algorithms, instead of “beating” existing offline/online methods. Instead, our algorithm can be added on top of them to further boost the performance.
>
>   Meanwhile, we conduct experiments to include more baselines and benchmark algorithms, including retrievers enhanced with rerankers, and three representative algorithms: DPR, ReFit and NUDGE. We present the result in Table 3. Overall, our method still keeps a competitive performance compared to previous SOTAs and features its deployment-time flexibility.
>
> - **Differentiation from Related Work**: We thank the reviewer for raising this point. Beyond the current discussion in the Appendix, we revise the statement of our introduction part to better demonstrate the novelties beyond the current literature:
>
>   *Existing deployments typically freeze embeddings and indices after offline training (Zeighami et al., 2024; Qin et al., 2023; Patil et al., 2024; Li et al., 2023), leaving no principled, low-cost way to repair performance at deployment. While recent work adapts controllers at inference time to decide when or how much to retrieve (Asai et al., 2024; Jeong et al., 2024) or tunes top-k and retrieval strategies (e.g., no retrieval / one-shot / multi-step) via multi-armed-bandit or reinforcement-learning approaches (Fu et al., 2024; Tang et al., 2024; Sun et al., 2025), these methods only adjust global hyperparameters but do not update the underlying embedding space and treat the underlying retriever system as fixed.*
>
>   *New problem formulation for online retrieval. We cast the problem of RAG tool and function selection under an online learning framework with bandit-style execution feedback (success/failure signals only for the chosen tool), updating the retrieval geometry on the fly after collecting each feedback. This new formulation itself gives a new perspective to understand RAG tasks.*
>
>   *A simple, scalable update rule. We propose an online gradient descent variant that adjusts embeddings per interaction using an importance-weighted estimator. The update of the embeddings keeps computation overhead minimal for large catalogs and high-throughput systems; it is both intuitive and theoretically supported. To the best of our knowledge, this is the first lightweight plug-and-play approach to improve retrieval performance at deployment time through adapting embeddings.*
>
> - **Sensitivity to Feedback**: For delayed feedback, the current algorithm can simply apply updates when feedback becomes available. The current analysis still applies with respect to the order in which updates are performed, i.e., a $O(\sqrt{T})$ regret.
>
>   As for noisy feedback, we first conduct experiments to check the influence of different noise levels on our algorithm performance. The experiment flips the binary feedback label in a Bernoulli manner, where each label samples a flip mask ~ Bernoulli(noise_rate) and gets flipped if the mask is 1. The sensitivity results are shown in Figure 11 of the revised version. Overall, our method is robust to the noisy labels and varying batch sizes. Meanwhile, for the noisy-label set up,  we extend our regret analysis to the noisy-label setting and show that the regret bound decomposes into a standard $O(\sqrt{T})$ term (as in the noiseless case) plus an additive $O(\rho T)$ term where $\rho$ is the noise rate (see in Appendix B.4).

---

### Official Review · Reviewer_iDTV · 2025-10-31

**Soundness:** 2
**Presentation:** 2
**Contribution:** 3
**Rating:** 4
**Confidence:** 3

**Summary:**

This paper proposes an online update strategy tailored for RAG with tool use and function calling. The high-level idea is that, at each request that comes at time step $t$, if the task was not correctly accomplished, a small gradient update is performed on the database/toolbase embeddings according to the back-propagation rule of the error. The paper has shown that the proposed paradigm is compatible with various aspects of a RAG system, including rerankers, time-varying databases, as well as multi-hop retrieval. Experimental results show that with approximately 50ms of extra latency, the method is able to achieve significant performance improvement over vanilla embedding-based retrieval, pushing beyond existing pareto frontier regarding the trade-off of retrieval latency and performance.

**Strengths:**

1. Given the minor latency introduced by the online update mechanism, the performance improvement achieved by the proposal is quite solid.
2. The paper did a good job to demonstrate the compatibility of the paradigm with commonly-used components of a modern RAG system.

**Weaknesses:**

1. I have some concerns regarding the comparisons made in the experiments -- there are existing work that conducts online updates of parameters for RAG, for example Fu et al. (2024) (https://arxiv.org/abs/2406.19251). Since none of the baselines incorporates any sort of test-time adaptive mechanism, I think some extra comparison along that line will be able to further strengthen the benefit of the proposal.
2. I have some issues with the presentation, listed below:

* 2a. L76 "requires no privileged access to model internals" -> I'm not sure if I fully understood what you mean here, but I think something like "can operate under a black-box setting" would be clearer.
* 2b. L107 reads a bit weird, here is what I think may work better: "In this section, we first develop the reader's intuition for our algorithm. We then introduce the rigorous terminology and notation used for its formal description."
* 2c. L169 "motivated by the sequentially arriving nature" of the user queries -> "motivated by the sequential arrival of user queries"
* 2d. Your Figure 4 and 5 should have roughly similar scale on the Y-axis for a clear visual understanding. For example, for left subfigure of Figure 5, the range of Y-axis is only 0.8%. I think this figure greatly exaggerated the impact of static vs. dynamic DB.
* 2e. L434 "The aim is not so much for a ..." -> colloquial language

**Questions:**

1. Can you talk a bit more about the "success criteria" for $1(i_t = i_t^*)$? I understand this could be user feedback or heuristic for real-world application deployment, but I'm not sure how they look like in your experiments. Are they also different for each of the test set?
2. I'm not sure if I follow the reason (b) spanning L177-179 -- why do you think that's might be the case for query embeddings but not for database/toolbase embeddings?

---

> ### Author Response · Authors · 2025-11-20
>
> We thank the reviewer for taking the time to read our paper and for all the comments. Our responses are as below:
>
> - **Comparison with Other Online Methods**: Regarding the comparison with existing online methods (e.g., Fu et al., 2024), we respectfully clarify the distinction between those approaches and ours. Methods such as AutoRAG-HP, MBA-RAG, and DynamicRAG utilize bandit or reinforcement learning techniques to adapt controller-level hyperparameters such as when to retrieve, the value of top-$k$, or which reranking policy to use. However, they treat the underlying retrieval system (the embedding space) as fixed. In contrast, ORAG adapts the retrieval geometry itself by updating item embeddings online. These two axes of adaptation are orthogonal and complementary rather than competitive:
>
>   - Controller methods decide *how* or *when* to retrieve.
>   - ORAG decides what implies similarity by mapping queries to items in the embedding space.
>
>   As noted in our introduction (Introduction, para 3), these methods can be seamlessly combined.
>
>   To further strengthen our empirical evaluation, we have expanded our experiments to include additional baselines, including retrievers enhanced with rerankers and three representative embedding optimization algorithms: DPR, ReFit, and NUDGE. As shown in Table 3, our method maintains competitive performance against these state-of-the-art approaches while uniquely featuring deployment-time flexibility.
>
> - **Presentation**: We appreciate the detailed editorial suggestions regarding the presentation (L76, L107, L169, Figure scales, etc.). We agree with these points and will incorporate all of them into the final revision to ensure clarity and consistency throughout the manuscript. We will also do a full pass to improve clarity and consistency throughout the paper.
> - **Feedback clarification**: Our binary feedback mechanism is designed to be both experimentally rigorous and practically deployable.
>   - In Standard Experiments (e.g., UltraTool): The datasets provide ground-truth labels. For example, for the query *"Set a reminder for the day before the meeting..."*, the dataset explicitly lists `set_reminder` as the correct tool. If the retriever selects this tool, the feedback is $1$ (success); otherwise, it is $0$ (failure).
>   - In Multi-Hop Experiments: Since these involve open-ended QA where exact matching is insufficient, correctness is evaluated using an LLM-as-a-judge.
>   - In Real-World Deployments: This setup extends naturally to practical applications. Simple tool-calling tasks can be verified by execution codes (e.g., HTTP 200 OK vs. error) or user feedback (thumbs-up/down), while complex tasks can be evaluated by a supervisor agent or model.
>
> - **Query fine-tuning clarification**: We thank the reviewer for raising this point. To clarify, a single query encoder is often shared across many applications and indices. Fine-tuning it for one RAG task risks degrading performance elsewhere. In contrast, tool/document embeddings are typically task- or application-specific; maintaining separate adapted copies for different applications or user segments is inexpensive and does not interfere with other systems.

---

> ### Comment · Reviewer_iDTV · 2025-11-27
>
> I thank the authors for the response. I read the updated draft, the other reviewer's reviews and the discussion. I noticed that in Table 3, the proposal only improves over existing baselines on one metric (R@10), but not the other (N@10), so in a way it actually confirms my concern rather than resolving it. I also wouldn't go so far to claim that R@10 "is particularly critical for real-world tool-use applications" -- RAG system deployment differs wildly based on their latency constraints and their underlying LLM context size and not all systems can afford to use 10 examples.
>
> I'm not convinced that the paper clears the acceptance threshold and hence will maintain my score.

---

> > ### Author Response · Authors · 2025-11-27
> >
> > We thank the reviewer for revisiting our updated experiments.
> >
> > **On R@10 vs N@10**. Our goal with Table 3 is not to claim that ORAG strictly dominates all strong baselines on every metric, but to show that a simple online embedding‑update rule can match the region of the latency–quality Pareto frontier reached by sophisticated offline training methods, while providing deployment‑time adaptivity. In Table 3, ORAG is competitive on N@10 with DPR/ReFit/NUDGE, while delivering consistent gains in R@10. This pattern is expected: ORAG primarily flips errors where the correct tool/document sits just outside the top‑k into the candidate set, which has a direct effect on recall but a smaller effect on a rank‑sensitive metric like N@10.
> >
> > **The selection of moderate k**. We agree that not all deployments can afford k=10. In our experiments, k=10 reflects a typical configuration where the LLM (or a reranker) considers a small candidate pool (for UltraTool, the whole pool size is ~2000) and then selects one or a few tools/passages. In such settings, the probability that the correct item appears anywhere in this pool, hence R@k is often the main bottleneck. We also respectfully argue that k=10 is a moderate choice, from both research and practical perspectives. For previous methods, they typically report main results on even larger k (DPR: top-20 and 100; Refit: top-100) and generally only report Recall. In the sense of practicality, LLMs with tool-calling ability are generally optimized for long-context dialogue, with context size usually 4k or higher. And the context length of 10 tools is affordable to them.
> >
> > **Relation to adaptive baselines**. Finally, we emphasize that the offline baselines in Table 3 require retraining the model parameters or performing offline deliberate tuning on task‑specific data, whereas ORAG operates purely at deployment time, requires no access to model internals, and can be layered on top of these methods. We therefore view the fact that ORAG is competitive in N@10 and improves R@10, while incurring only ~50ms extra latency and no retraining cost, as evidence that online embedding adaptation is a practically useful and complementary capability rather than a direct competitor to existing training‑time methods.

---

### Official Review · Reviewer_EW7w · 2025-11-01

**Soundness:** 3
**Presentation:** 4
**Contribution:** 3
**Rating:** 6
**Confidence:** 4

**Summary:**

This paper proposes an online learning framework that updates retrieval embeddings during deployment to improve tool selection in RAG.

**Strengths:**

1. The approach is lightweight and avoids LLM finetuning.
2. The plug-and-play online embedding update mechanism requires only binary signals.
3. The effectiveness of the proposed method is supported by empirical evaluations across multiple datasets, and it shows consistent improvements on Recall@10 and task success.
4. The paper is clearly written, grounded in online convex optimization theory, providing rigor beyond empirical work.

**Weaknesses:**

1. It would be useful to analyze robustness to continuous signals.
2. Continual online updates may gradually distort embedding space, more discussions on its stabilization would be helpful.
3. A deeper analysis on memory for large tool catalogs, and effect of update frequency are lacking.

**Questions:**

1. Is there catastrophic drift during online training? Are resets or replay buffers needed?
2. How does performance compare to offline embedding finetuning methods such as RAFT?
3. How sensitive is performance to the learning rate schedule?
4. What happens under cold-start / unseen tools where p is near zero?

---

> ### Author Response · Authors · 2025-11-20
> **Official Comment by Authors (Part 1)**
>
> We thank the reviewer for taking the time to read our paper and for all the comments. Our responses are as below:
>
> - **Continuous signals**: Our current design prioritizes binary success/failure feedback because such signals are common in real-world systems (e.g., thumbs up/down, pass/fail unit tests, or rule-based verifiers) and are cost-effective to collect at scale. Regarding "continuous signals", we interpret this as potentially referring to scalar feedback (such as relevance scores or confidence margins). Our gradient estimator can be theoretically extended to weighted feedback by directly using the continuous score to construct a gradient for some given loss function: since the score is explicitly given and there is no bandit feedback, we can directly calculate the gradient given a loss function. However, to ensure we fully address the reviewer's specific concern, could the reviewer kindly specify an example of the continuous signal envisioned? We would be happy to discuss this further during the discussion period.
>
> - **Embedding space stability and catastrophic drift**: We appreciate the reviewer raising the issue of potential embedding distortion. In a theoretical worst-case scenario where the system repeatedly receives the exact same query and reinforces the same tool indefinitely, the selected tool's embedding would be pulled toward the query while others are pushed away, potentially polarizing the space (as illustrated in Figure 2). However, our empirical results show that this does not lead to instability in practice for several reasons:
>
>   - Diverse Query Distribution: Real-world RAG systems process a diverse distribution of queries rather than repeating a single instance, preventing the collapse of the embedding geometry.
>   - Adaptation to Shift: If a distribution shift occurs over time, our algorithm is explicitly designed to adapt the retriever to this new distribution. While a temporary "drift" from the old distribution is expected, it represents a desirable adaptation to the new task distribution.
>   - Gradual Updates: We utilize small learning rates and an update rule grounded in online convex optimization, which encourages gradual rather than drastic changes. As detailed in our theoretical analysis (Theorem 5.1), the cumulative regret grows at most $O(\sqrt{T})$, ensuring the average loss converges to the optimum rather than diverging.
>
>   To empirically verify stability, we conducted new experiments tracking performance on the "first 10 seen queries" throughout the training process. As shown in Figure 9, while minor fluctuations occur, there is no sign of performance degradation or catastrophic forgetting over time.
>
> - **Memory/update frequency**: Our algorithm is memory-efficient and does not require a replay buffer or additional memory beyond the standard embedding index. We perform in-place updates to the existing item embedding matrix via simple addition and subtraction. For very large catalogs, we describe a variant in Appendix B.2 that updates only the sampled item, reducing per-step computational complexity to $O(d)$ (where $d$ is the embedding dimension), making it independent of the total number of tools.
>
>   Regarding update frequency, our algorithm naturally supports mini-batches, allowing gradients to be accumulated over multiple interactions to reduce update frequency. This trades adaptation speed for smoother updates. We have added a sensitivity analysis regarding batch sizes in Figure 11. The results demonstrate that our method remains robust across varying batch sizes.

---

> ### Author Response · Authors · 2025-11-20
> **Official Comment by Authors (Part 2)**
>
> > (Continued)
>
> - **New benchmarks**:  We want to first respectfully clarify that our algorithm is a lightweight deployment-time adaptation of general base retrievers. It is plug-and-play and can be applied to any retriever (and is compatible with other offline training methods). Our experiments are designed to show our algorithm’s effectiveness across retrievers under different situations (e.g., different datasets and base models, single-hop and multi-hop pipelines, etc). Thus, we compare the performance with and without applying our algorithm of the same base models. The reason we include more benchmarks is to show that, even the original retriever has a worse performance than some benchmarks (e.g., bge-reranker-v2-gemma), after applying our methods, the performance can be comparable across different datasets (and with less retrieval time), as shown in Table 1.
>
>   For new benchmarks, we have added performance comparisons for offline fine-tuning baselines, specifically DPR, ReFit, and NUDGE, in Table 3. While we understand the importance of methods like RAFT, we note that RAFT primarily targets the "Generation" aspect of RAG (teaching the LLM to distinguish documents and ignore distractors), whereas our method focuses strictly on optimizing the "Retrieval" geometry. Therefore, we selected baselines that specifically target embedding optimization to add:
>
>   - DPR: Fine-tunes the embedding model itself.
>
>   - ReFit/NUDGE: Optimize query or document embeddings offline.
>
>   As shown in Table 3, our method maintains competitive performance against these SOTA offline methods while offering the unique advantage of deployment-time flexibility without requiring annotated training data beforehand.
>
> - **Sensitivity to Learning rate**: We appreciate the reviewer for pointing this and we include a new ablation plotting performance under several learning-rate schedules: (i) Constant, the learning rate remains unchanged during the training process; (ii) Inverse-square-root decay, where the learning rate decays with steps t by $\sqrt{t}$; (iii) Cosine decay, where the learning rate with cosine curve, i.e., decreasing slow first then faster in the middle, and slow again when lr is small. We keep the smallest lr as 0.05 * initial lr. The results are shown in Figure 10. In all decaying cases, the method demonstrates robust convergence.
>
> - **Cold-start / unseen tools**: We first clarify that when a new tool is added with a reasonable initial embedding (e.g., computed from its documentation), its selection probability need not be near zero: the retriever is not cold-started and can even assign a large probability if this new tool is matched with the query based on its documentation. Our dynamic-database experiment (Figure 5) shows that when previously missing optimal tools are introduced, ORAG adapts by pushing earlier suboptimal items away from relevant queries, making room for the new tool to be selected more often even before its own updates. One extreme case is that the new tool has no documentation, and the retriever cannot assign any probability (or a zero probability). In this case, one solution is to have some exploration probability (e.g., $10^{-2}$) to retrieve this new tool for several periods to initialize its embeddings.

---

> > ### Comment · Reviewer_EW7w · 2025-11-20
> >
> > Thanks for the response and for addressing my concerns. I maintain my overall evaluation and positive score.

---

### Official Review · Reviewer_SoPM · 2025-11-03

**Soundness:** 2
**Presentation:** 3
**Contribution:** 2
**Rating:** 4
**Confidence:** 4

**Summary:**

The paper proposes Online-Optimized RAG, a deployment-time procedure that updates tool embeddings online from minimal feedback (task success/failure), without touching the LLM. The retriever is modeled as a softmax over inner products; after each interaction, the method applies a lightweight online gradient update to the item embeddings, pushing the selected tool toward the query if it succeeded and away if it failed, while pushing unselected items away due to the softmax coupling.

**Strengths:**

1. Updating only the item embeddings from success/failure, no LLM changes; works for tools and general document retrieval.

2. Stochastic gradient view with an unbiased estimator and a standard sublinear regret analysis.

**Weaknesses:**

1. The novelty is limited. The formulation (softmax retriever + online gradient with bandit feedback) is incremental relative to known online retrieval/adaptation ideas.

2. Main comparisons are dense encoders, BM25, and two rerankers; missing hybrid/metadata recall, supervised tool retrievers, learned rerankers, and online alternatives (e.g., query-side adapters).

3. The paper doesn’t report significance, and there’s little analysis of sensitivity to learning-rate or resilience to noisy/incorrect feedback.

4. When a reranker is used, it’s ambiguous how much of a success is due to the sampler’s updated embeddings vs. the reranker’s scoring. The current estimator doesn’t separate these effects, so the online updates may be biased toward whatever the reranker already prefers.

**Questions:**

1. Can you add non-encoder baselines for tool recall (e.g., BM25/hybrid with metadata,  supervised dense retriever + reranker, a tag/graph-aware recall)?

2. Include at least one online alternative (e.g., a query-side online adapter) in the tool-use setting.

3. When a reranker is used, how do you attribute success to the sampler vs. the reranker? Any bias correction beyond the current estimator?

---

> ### Author Response · Authors · 2025-11-20
> **Official Comment by Authors (Part 1)**
>
> We thank the reviewer for taking the time to read our paper and for all the comments. Our responses are as below:
>
> - **Novelty (weakness 1)**: We kindly argue that our novelty is significant to the literature. As summarized in the introduction part, for the current literature
>
>   *“While recent work adapts controllers at inference time to decide when or how much to retrieve (Asai et al., 2024; Jeong et al., 2024) or tunes top-k and retrieval strategies (e.g., no retrieval/one-shot /multi-step) via multi-armed-bandit or reinforcement-learning approaches (Fu et al., 2024; Tang et al., 2024; Sun et al., 2025), these methods only adjust global hyperparameters but do not update the underlying embedding space.”.*
>
>   To the best of our knowledge: (i) we are the first to cast tool/document retrieval as online multiclass prediction under bandit feedback, and develop unbiased gradient estimators tailored to softmax retrievers. This new formulation itself gives a new perspective to understand RAG tasks. (ii) Our proposed algorithm is the first lightweight plug-and-play procedure to improve retrieval performance at deployment time. It only uses binary success/failure signals, requires no changes or access to LLM internals, and operates under a black-box setting, and works with dynamic catalogs, multi-hop pipelines, and rerankers. (iii) We further provide analyses that explicitly show how the algorithm’s performance depends on the initialization quality and the confidence in the optimal item, giving actionable insight into when deployment-time adaptation is most beneficial. All these novelties are beyond the standard ingredients (softmax retriever + online gradient with bandit feedback) in the current literature.
>
> - **Comparisons and baseline (weakness 2 and questions 1)**: We want to kindly clarify that our algorithm is a lightweight deployment-time adaptation of general fully trained base retrievers to improve their performance. It is plug-and-play and can be applied to any retriever. And our experiments are designed to show our algorithm’s effectiveness across retrievers under different situations (e.g., different datasets and base models, single-hop and multi-hop pipelines, etc). Thus, we compare the performance with and without applying our algorithm to the same base models. The reason we include more benchmarks is to show that, even the original retriever has a worse performance than some benchmarks (e.g., bge-reranker-v2-gemma), after applying our methods, the performance can be comparable across different datasets (and with less retrieval time), as shown in Table 1.
>
>   To further enhance the comparison, we add more baselines: DPR, ReFit, and NUDGE. These methods are representative methods for improving tool call performance, and each of them represents a different methodology. DPR tries to directly improve the embedding model itself. ReFit tries to optimize on the query embedding side, and NUDGE also aims to optimize the document embeddings, but in a fully offline manner. We present the result in Table 3 in the revised PDF. Overall, our method still keeps a competitive performance compared to previous SOTAs and features its deployment-time flexibility.

---

> ### Author Response · Authors · 2025-11-20
> **Official Comment by Authors (Part 2)**
>
> > (Continued)
>
> - **Reported metrics and ablation study (weakness 3)**: We thank the reviewer for pointing this out. For experiments with multiple runs, we have added the standard error for better reference in the revised version. We have also added sensitivity tests for learning rate, batch size, and noise rate. The noise rate experiment flips the binary feedback label in a Bernoulli manner, where each label samples a flip mask ~ Bernoulli(noise_rate) and gets flipped if the mask is 1. The sensitivity results are shown in Figure 10 and 11. Overall, our method is robust to the noisy labels and varying batch sizes.
>
>   Meanwhile, for the noisy-label set up, in Appendix B.4  we extend our regret analysis to the noisy-label setting and show that the regret bound decomposes into a standard $O(\sqrt{T})$ term (as in the noiseless case) plus an additive $O(\rho T)$ term where $\rho$ is the probability of (randomly) flipping the feedback.
>
> - **Contribution of reranker and its bias correction (weakness 4 and question 3)**: We thank the reviewer for the question. We kindly clarify that our algorithm is plug-and-play for any retriever with or without a reranker. When a reranker is used, our objective is to optimize the end-to-end success of the whole pipeline. Thus, there is no competition relationship between the retriever and the reranker - they are compatible, and we can always apply both of them in practice. Meanwhile, to better showcase the performance under different settings and separate the contributions of the reranker and our method, we add two baselines where the vanilla/base retriever and our method are respectively equipped with a reranker, as shown in Figure 12. The results show that our method can consistently improve the performance of the vanilla/base retriever under both settings (with and without a reranker).
>
>   Additionally, we agree that with a reranker, the current gradient estimator is no longer an unbiased estimator of the pure softmax retriever loss. In principle, if the reranker’s selection probabilities were available, one could similarly construct an unbiased estimator by correcting the importance weighting with the final choice probability. However, in practice, as shown in Figure 4, we observe that even without such correctness, the current biased estimator still works well: if the reranker’s final choice has a small retriever’s output probability, the bias is larger, and this small probability will contribute a potentially significant update to the embeddings and improve future retrieval performances. On the other hand, if the probability is large, then the bias is small and such bias may not cause a big impact on the update.
>
> - **Online alternative / a query-side online adapter (question 2)**: (i) To the best of our knowledge, previous online RAG methods such as AutoRAG-HP, MBA-RAG, and DynamicRAG adapt global hyperparameters (e.g., when or how much to retrieve) instead of the retrieval step itself. For example, there is no online alternative that can be applied to a single-hop setup. The most similar method is ReFit, an offline method to adjust the query embeddings, and its baseline has been implemented and compared in Table 3. (ii) Meanwhile, for a query-side online adapter, as discussed in our work, we don’t consider it practical for two reasons: *“(a) the query embedding models are sometimes blackbox APIs and don’t provide a fine-tuning option, and (b) they are often used simultaneously for other RAG tasks, and fine-tuning against one RAG task may deteriorate its performances on others.”*

---

### Official Review · Reviewer_6KJn · 2025-11-04

**Soundness:** 3
**Presentation:** 3
**Contribution:** 2
**Rating:** 4
**Confidence:** 4

**Summary:**

This paper proposes Online-Optimized RAG, a framework that continuously adapts retrieval embeddings during deployment using minimal feedback (success/failure signals). The method treats tool retrieval as online multi-class classification and applies lightweight gradient updates to tool embeddings after each interaction. While the motivation is compelling and the approach theoretically sound, I have significant concerns about the experimental evaluation (see below).

**Strengths:**

- Well-motivated problem: Addresses a genuine practical challenge where embedding models become misaligned with operational environments due to outdated models, or distribution shifts.
- Practical approach: The method requires only binary feedback (success/failure) and adds minimal computational overhead.
Solid theoretical foundation
- Broad applicability: Demonstrates extensions to K-retrieval with reranking, dynamic databases, and multi-hop scenarios

**Weaknesses:**

- Unclear experimental evaluation: The primary concern is whether the method learns from test data while baselines remain static. The paper reports results "after applying the method with an average of 3000 updates" - but it's unclear if these updates come from test interactions, creating an unfair advantage over baseline embeddings. There appears to be no clear temporal split where the method trains on earlier interactions and is evaluated on later ones. This is critical for validating online learning approaches and ensuring fair comparison.
- Baselines: The comparison is primarily against static embedding models that cannot adapt. Comparisons could include: Other online learning methods, oracle baselines trained offline on the same feedback data, or other adaptive versions of the baseline embedding models
- Limited realistic evaluation: While Table 2 shows "single exposure" results with modest improvements (1-2%), the main results rely on "multiple exposure" settings that may not reflect realistic deployment scenarios.

**Questions:**

Do you perform temporal evaluation where the method learns from earlier time periods and is tested on later periods? If not, how do you ensure fair comparison when the method adapts to the test distribution?

When you report performance metrics, are the embedding updates derived from the same queries being evaluated?

Overall, the details about the feedback collection process should be clearer.

---

> ### Author Response · Authors · 2025-11-20
> **Official Comment by Authors (Part 1)**
>
> We thank the reviewer for taking the time to read our paper and for all the comments.
>
> Regarding the primary concern about “whether the method learns from test data while baselines remain static”, we wish to clarify that this capability, i.e., adapting during deployment, is the core contribution of our proposed algorithm. Specifically, recall our algorithm is a lightweight deployment-time adaptation of general fully trained base retrievers to improve their performance. From a point of the standard RAG, the deployment phase is the testing phase, and every deployment query is a testing query. Thus, we believe it is fair to evaluate and compare our updated retrievers with the static retrievers - both are evaluated on the same deployment/testing queries, and our algorithm never requires a ground-truth label during the deployment. From another perspective of online learning, for each new arrival query, our algorithm selects which tool to retrieve, and then updates the tool embeddings based on the feedback. The decision for each new arrival query only depends on past interactions. In this sense, each new query constitutes a temporal split and is a testing sample, and all previous interactions act as “training samples”. In contrast, the static baselines just don’t utilize all these “training samples”. Thus, it is also reasonable and fair to compare our algorithm with standard static retrievers.

---

> ### Author Response · Authors · 2025-11-20
> **Official Comment by Authors (Part 2)**
>
> We further want to make the following specific clarifications for our evaluation:
>
> - **Fairness and temporal evaluation**. Each query’s true label is never directly revealed to the algorithm. Even under multiple exposure cases, there is no leakage of the testing samples (for each new arrival query), and the updates come from the training samples (previous queries). Thus, all our results are such temporal evaluations “where the method learns from earlier time periods and is tested on later periods”. To better check the generalizability of our algorithm, we added experiments to test its performance on the never-seen queries. Specifically, we split the original queries into training and test sets, and only train the algorithm on the training queries and evaluate its performance on the test queries (which are never seen during the updates). The results are shown in Figure 9 of the revised PDF and demonstrate a good generalizability over the unseen queries.
> - **Evaluation and the source of updates**. The updates are from the training data (queries seen before the evaluation), and the results are evaluated on the queries. Before the revision, for some experiments (e.g., Table 1), while the test queries can be seen before from the history data, their true labels are never directly revealed to the algorithm and are only accessible through bandit feedback.
> - **Baselines**. We want to first kindly clarify that other online methods, such as AutoRAG-HP, MBA-RAG, and DynamicRAG, adapt controller-level hyperparameters (e.g., when to retrieve, top-k, or reranking policies) using bandit or RL techniques, but they treat the underlying retriever system as fixed. ORAG instead adapts the retrieval geometry itself by updating item embeddings online. These axes of adaptation are complementary instead of competitors and can be seamlessly combined together: (i) Controller methods decide how much or when to retrieve; (ii) ORAG decides how queries map to items in the embedding space. Meanwhile, to ensure a more comprehensive comparison, we add several classic or the latest techniques addressing retrieval misalignment problems. We present the result in Table 3 in the revised PDF. Overall, our method still keeps a competitive performance compared to previous SOTAs and features its deployment-time flexibility.
>
> - **Single vs. Multiple Exposure**. We fully agree that single-exposure evaluation is important, which is why we already include single-exposure results in Table 2. At the same time, multiple exposures are common in real tool-use deployments: (i) Users often issue follow-up or paraphrased queries for the same tool. (ii) Modern agentic pipelines use planners and query rewriters that generate several closely related sub-queries for a given tool. For example, CoRAG [1] generates multiple retrieval chains from one given query and adds the successful ones to the training set for later finetuning; SQL-Genie [2] maintains an “example bank” to retrieve a similar past query and "reuse" its SQL structure when possible; MaFeRw [3] tries to rewrite queries to be similar enough for retrieving the same document to improve downstream performance. Our multiple-exposure setting is designed to approximate this realistic “clustered” usage pattern. ORAG shows consistent gains in both regimes, with larger improvements under multiple exposures and smaller but still positive gains when each query is seen only once. This is also a common practice in offline methods like DPR and NUDGE, which require multiple gradient update steps on the training data.
>
> - **Feedback collection**. In a real-world setting, the binary feedback can be obtained by the user feedback (like a thumb-up or thumb-down), or using a lightweight LLM-as-a-judge to evaluate whether the user’s queries are addressed by the retrieved tool/item. In our experiments, we use the ground-truth label from the datasets to act as a “verifier”: the dataset offers the ground-truth optimal tool/item for each query. Thus, in the experiments, if the algorithm’s output tool/item is the ground-truth one, it receives the signal one; otherwise, the signal is zero.
>
> [1] Wang, L., Chen, H., Yang, N., Huang, X., Dou, Z., & Wei, F. (2025). Chain-of-Retrieval Augmented Generation. *ArXiv*. https://arxiv.org/abs/2501.14342
>
> [2] Ghosh, P., Jain, A., & Yenigalla, P. (2025). SQLGenie: A practical LLM based system for reliable and efficient SQL generation. ACL 2025. https://aclanthology.org/2025.acl-industry.71
>
> [3] Wang, Y., Zhang, H., Pang, L., Guo, B., Zheng, H., & Zheng, Z. (2024). MaFeRw: Query Rewriting with Multi-Aspect Feedbacks for Retrieval-Augmented Large Language Models. *ArXiv*. https://arxiv.org/abs/2408.17072

---

### Author Response · Authors · 2025-11-20
**Summary of Revisions and Response to Common Concerns**

We thank all the reviewers for your detailed suggestions and comments. We appreciate the time spent by the reviewers, especially given this year’s review workload. Below we provide a summary of the main clarifications, additional experiments, and revisions we have made.

We first clarify the key contributions of this work to the RAG literature. (1) Methodologically, we cast tool/document retrieval as an online multiclass prediction problem under bandit feedback and derive unbiased gradient estimators tailored to softmax-based retrievers, providing a new lens on RAG and tool use. (2) Practically, our algorithm ORAG is the first lightweight, plug‑and‑play deployment‑time embedding adapter: it updates item embeddings online using only binary success/failure signals, requires no access to LLM internals, and can operate under a black-box setting, and works with arbitrary base retrievers, dynamic catalogs, multi‑hop pipelines, and rerankers. (3) Theoretically, we give regret guarantees that explicitly characterize how performance depends on initialization and confidence in the optimal item. In the revision, we also extend this analysis to noisy feedback.

Below, we summarize how we address the common concerns raised by the reviewers:

- Evaluation and fairness. Our method is a deployment-time adapter that updates the retriever only from past interactions and never directly observes the true label (optimal item) for the current query, so each new query is effectively a fresh test point while static baselines simply ignore history. This makes it reasonable to compare an adapted retriever (e.g., ORAG) against static ones on the same deployment stream, because both are evaluated on exactly the same test queries. To further address the concern about “learning from test data”, we add a standard temporal hold-out experiment where ORAG is evaluated on a disjoint stream of never-seen queries (See Appendix C.5.1 Figure 9.(a)). These results clearly show that the gains from ORAG transfer to unseen queries and do not rely on repeatedly tuning to the same queries. For the embedding stability, we track performance over time on all seen queries and specifically on the first few seen queries (See Appendix C.5.1 Figure 9. (b) (c)), and we observe no catastrophic forgetting or sudden performance drops, supporting the stability of the online updates.

- Comparisons and positioning. ORAG is designed as a lightweight deployment-time adapter that can be stacked on top of existing retrievers and controllers rather than as a direct competitor to them. As discussed in the paper, previous online RAG methods such as AutoRAG-HP, MBA-RAG, and DynamicRAG adapt global hyperparameters (e.g., when or how much to retrieve, or top‑k) while treating the embedding space as fixed, whereas ORAG directly adapts the retrieval geometry by updating item embeddings online. These two axes are complementary: controllers decide whether and when to retrieve, while ORAG decides which items are favored for a given query, and they can be used together. Similarly, generation-focused methods such as RAFT primarily improve how the generator handles retrieved documents and distractors, whereas ORAG focuses on improving the retrieval component itself. To strengthen empirical comparisons suggested by the reviewers, we include three representative fine-tuning baselines (DPR, ReFit, and NUDGE) as well as retriever+reranker pipelines where both the vanilla and ORAG-adapted retrievers are paired with the same reranker (see Appendix C.5.3). Across datasets, ORAG remains competitive with these stronger baselines and reliably improves its base retrievers, while preserving its advantages of no retriever retraining, compatibility with dynamic catalogs, and minimal deployment overhead.

- Ablations, robustness, and noisy feedback. Following the reviews’ suggestions, we expand the empirical analysis with sensitivity tests over learning rate (schedule), batch size, and the amount of label noise (see Appendix C.5.2). For batch size, ORAG naturally supports mini-batch updates, and we find that varying the update frequency trades off adaptation speed against smoother updates. For noisy feedback, we explicitly flip binary labels with different noise rates and observe that ORAG remains robust under moderate noise, maintaining clear gains over the base retrievers. Additionally, for experiments with multiple runs, we now add standard errors, providing a clearer picture of variability in the reported metrics. On the theory side, we extend our regret analysis to the noisy-label setting and show that the regret bound decomposes into a standard $O(\sqrt{T})$ term (as in the noiseless case) plus an additive $O(\rho T)$ term where $\rho$ is the probability of (randomly) flipping the feedback (see Appendix B.4). Together, these empirical and theoretical results indicate that ORAG is stable across a broad range of hyperparameters and feedback conditions.

---

### Meta-Review · Area_Chair_dd7b · 2025-12-09

**Summary:**

This work seeks to address a real deployment problem, namely embedding drift, with a lightweight, plug-and-play online update that uses only binary success/failure feedback, adds ~tens of ms latency, and leaves the LLM untouched. Comes with a simple softmax-bandit formulation, regret grounding, and shows consistent gains across settings including K-retrieval with reranking, dynamic databases, and multi-hop. Meanwhile, the reviewers are concerned that the evaluation fairness is unclear (possible learning on test interactions, no temporal split), and baselines omit online/adaptive and fine-tuning alternatives needed to contextualize gains. Statistical rigor and sensitivity analyses are thin, attribution is muddied when rerankers are used, and robustness to noisy or delayed feedback and long-run drift is underexplored.

**Reviewer Concerns:**

Addressed: The rebuttal adds a temporal hold-out evaluation and shows stability over time, clarifying fairness and alleviating “learning on test data” concerns. It strengthens comparisons by adding fine-tuning baselines (DPR, ReFit, NUDGE) and matched reranker pipelines, and clarifies positioning relative to controller-style online RAG. Sensitivity studies over learning rate, batch size, and label-noise were included, with extended regret analysis for noisy feedback and reporting of variability.

Outstanding: External validity would benefit from larger-scale, longer-horizon deployments with hardware-normalized latency and memory profiles, plus delayed or adversarial feedback stress tests. Stronger head-to-head against online/adaptive alternatives (e.g., query-side adapters, DynamicRAG-style methods) and clearer attribution when rerankers are used remain desirable. Further analysis of cold-start, unseen tools, and drift under evolving catalogs would round out the claims.

**Reviewer Scores:**

H1oD may be changed from 4 to 6 as the requested new results were provided. The others might be unchanged.

---

### Decision · Program_Chairs · 2026-01-26

Reject